# NO$_2$-initiated multiphase oxidation of SO$_2$ by O$_2$ on CaCO$_3$ particles

Ting Yu[*], Defeng Zhao[*, a], Xiaojuan Song, Tong Zhu

BIC-ESAT and SKL-ESPC, College of Environmental Sciences and Engineering, Peking University, Beijing, 100871, China

[*]These authors contributed equally to this work.

[a]Now at: Department of Atmospheric and Oceanic Sciences & Institute of Atmospheric Sciences, Fudan University, Shanghai, 200438, China

*Correspondence to*: Tong Zhu (tzhu@pku.edu.cn)

**Abstract.** The reaction of SO$_2$ with NO$_2$ on the surface of aerosol particles has been suggested to be important in sulfate formation during severe air pollution episodes in China. However, we found that the direct oxidation of SO$_2$ by NO$_2$ was slow and might not be the main reason for sulfate formation in ambient air. In this study, we investigated the multiphase reaction of SO$_2$ with an O$_2$/NO$_2$ mixture on single CaCO$_3$ particles using Micro-Raman spectroscopy. The reaction converted the CaCO$_3$ particle to a Ca(NO$_3$)$_2$ droplet, with CaSO$_4$•2H$_2$O solid particles embedded in it, which constituted a significant fraction of the droplet volume at the end of the reaction. The reactive uptake coefficient of SO$_2$ for sulfate formation was on the order of $10^{-5}$, which was higher than that for the multiphase reaction of SO$_2$ directly with NO$_2$ by 2–3 orders of magnitude. According to our observations and the literature, we found that in the multiphase reaction of SO$_2$ with the O$_2$/NO$_2$ mixture, O$_2$ was the main oxidant of SO$_2$ and was necessary for radical chain propagation. NO$_2$ acted as the initiator of radical formation, but not as the main oxidant. The synergy of NO$_2$ and O$_2$ resulted in much faster sulfate formation than the sum of the reaction rates with NO$_2$ and with O$_2$ alone. We estimated that the multiphase oxidation of SO$_2$ by O$_2$ initiated by NO$_2$ could be an important source of sulfate and a sink of SO$_2$, based on the calculated lifetime of SO$_2$ regarding the loss through the multiphase reaction versus the loss through the gas-phase reaction with OH radical. Parameterizing the reactive uptake coefficient of the reaction observed in our laboratory for further model simulation is needed, as well as an integrated assessment based on field observations, laboratory study results, and model simulations to evaluate the importance of the reaction in ambient air during severe air pollution episodes, especially in China.

## 1 Introduction

It has been suggested that multiphase or heterogeneous oxidation of $SO_2$ potentially plays an important role in sulfate formation in the atmosphere (Seinfeld and Pandis, 2006). During the severe pollution episodes that occur frequently in China, high sulfate concentrations cannot be explained by the gas phase oxidation of $SO_2$ and its well-known aqueous chemistry (Zheng et al., 2015a; Cheng et al., 2016), highlighting the role of under-appreciated heterogeneous oxidation or multiphase pathways.

Recently, the multiphase oxidation of $SO_2$ by $NO_2$ has been introduced in air quality model simulations to explain the discrepancy between the modeled and observed sulfate concentration during severe pollution episodes in China (Cheng et al., 2016; Gao et al., 2016; Wang et al., 2016; Xue et al., 2016), despite the uncertainties in the kinetic parameters for $SO_2$ oxidation and in the pH value of aerosol particles in China (Wang et al., 2016; Cheng et al., 2016; Liu et al., 2017; Guo et al., 2017). However, according to our recently published results (Zhao et al., 2017), the direct oxidation of $SO_2$ by $NO_2$ could not contribute significantly to sulfate formation in the atmosphere because the reactive uptake coefficient of $SO_2$ for sulfate formation due to direct oxidation by $NO_2$ is very low ($\sim 10^{-8}$).

Although the contribution of the direct oxidation of $SO_2$ by $NO_2$ to sulfate formation is not significant, $NO_2$ may be involved in other oxidation pathways of $SO_2$. It has been reported that the reaction of $NO_2$ with $SO_3^{2-}$ and $HSO_3^-$ in the bulk aqueous phase can form the $SO_3^{\bullet-}$ radical, which can further react with $O_2$ and produce a series of radicals that oxidize S(IV) species (Littlejohn et al., 1993). The reaction pathway may result in a fast $SO_2$ oxidation due to the potential synergy of $NO_2$ and $O_2$.

Despite such a reaction mechanism for $SO_2$ oxidation being proposed, its role in $SO_2$ oxidation in the ambient atmosphere is not well established. Most previous studies have focused on the direct reaction of $SO_2$ with $NO_2$, including the determination of its rate constant (Lee and Schwartz, 1983; Clifton et al., 1988; Shen and Rochelle, 1998; Spindler et al., 2003; Nash, 1979; Huie and Neta, 1986). According to the reaction products and their reported yields (Lee and Schwartz, 1983; Clifton et al., 1988), the overall reaction equations of the direct reaction of $SO_2$ with $NO_2$ are as follows:

$$2NO_2(aq) + HSO_3^-(aq) + H_2O \rightarrow 2NO_2^-(aq) + SO_4^{2-}(aq) + 3H^+(aq), \tag{R1}$$

$$2NO_2(aq) + SO_3^{2-}(aq) + H_2O \rightarrow 2NO_2^-(aq) + SO_4^{2-}(aq) + 2H^+(aq), \tag{R2}$$

and the reactions are proposed to proceed via $NO_2$–S(IV) adduct complexes (Clifton et al., 1988).

$$NO_2(aq) + SO_3^{2-}(aq) \rightarrow [NO_2 - SO_3]^{2-}(aq). \tag{R3}$$

$$NO_2(aq) + [NO_2 - SO_3]^{2-}(aq) \rightarrow [NO_2 - SO_3 - NO_2]^{2-}(aq). \tag{R4}$$

$$[NO_2 - SO_3 - NO_2]^{2-}(aq) + OH^-(aq) \rightarrow [NO_2 - SO_4H - NO_2]^{3-}(aq). \tag{R5}$$

$$[NO_2 - SO_4H - NO_2]^{3-}(aq) \rightarrow 2NO_2^-(aq) + SO_4^{2-}(aq) + H^+(aq). \tag{R6}$$

Additionally, $NO_2$–S(IV) adduct complex may decompose as follows (Spindler et al., 2003).

$$[NO_2 - SO_3]^{2-}(aq) \rightarrow NO_2^-(aq) + SO_3^{\bullet-}(aq). \tag{R7}$$

However, studies of the oxidation rate of $SO_2$ at the $O_2$ concentrations relevant to the ambient atmosphere and the potential influence of the synergy of $NO_2$ and $O_2$ on the oxidation rate are very limited (Turšič et al., 2001; He et al., 2014), except a few studies investigated $SO_2$ oxidation in the presence of $NO_2$ as well as $O_2$ (Littlejohn et al., 1993; Shen and Rochelle, 1998; Santachiara et al., 1990). Moreover, previous studies have mainly focused on the

reaction in bulk solution and only few studies have investigated the oxidation of $SO_2$ by $NO_2$ on aerosol particles (Santachiara et al., 1990, 1993). On aerosol particles, water activity, pH, ionic strength, the presence of other compounds or ions, and the role of particle surface are different from in dilute bulk solution and may affect the reaction process and reaction rate. Therefore, further studies of the multiphase reaction of $SO_2$ with $O_2/NO_2$ mixtures on aerosol particles are required to determine the kinetic parameters and the mechanism of the reaction.

In this study, we investigated the multiphase reaction of $SO_2$ with $O_2$ in the presence of $NO_2$ on $CaCO_3$ particles. We quantified the reactive uptake coefficient of $SO_2$ due to the reaction with an $O_2/NO_2/H_2O$ mixture. Based on our observations and the existing literature, we further discussed the reaction mechanism. Furthermore, we estimated the role of the multiphase oxidation of $SO_2$ by $O_2$ in the presence of $NO_2$ in the atmosphere.

## 2  Experimental

The experiments were conducted using a flow reaction system and the setup is shown in Fig. S1. The experimental setup and procedure used have been described in detail in previous studies (Zhao et al., 2017; Zhao et al., 2011; Liu et al., 2008). A gas mixture of $NO_2$, $SO_2$, $O_2$, $N_2$, and water vapor reacted with particles deposited on a substrate in the flow reaction cell. The concentrations of $SO_2$ and $NO_2$ were controlled using mass flow controllers by varying the flow rates of $SO_2$ (2,000 ppm in high purity $N_2$, National Institute of Metrology P.R. China), $NO_2$ (1,000 ppm in high purity $N_2$, Messer, Germany), and synthetic air [20% $O_2$ (high purity grade: 99.999%, Beijing Haikeyuanchang Practical Gas Co., Ltd.) and 80% $N_2$ (high purity grade: 99.999%, Beijing Haikeyuanchang Practical Gas Co., Ltd.)]. Relative humidity (RH) was controlled by regulating the flow rates of reactant gases, dry synthetic air, and humidified synthetic air. Humidified synthetic air was prepared by bubbling synthetic air through fritted glass in water. In some experiments, the $O_2$ concentrations were varied by regulating the mixing ratios of $O_2$ and $N_2$ to investigate the effect of $O_2$. $SO_2/O_2/NO_2/H_2O$ mixtures flew through the reaction cell and reacted with individual stationary $CaCO_3$ particles, which were deposited on a Teflon-FEP film substrate annealed to a silicon wafer. RH and temperature were measured using a hygrometer (HMT100, Vaisala, Vantaa, Finland) at the exit of the reaction cell. Additionally, temperature was measured using another small temperature sensor (Pt 100, 1/3 DIN B, Heraeus, Hanau, Germany) in the reaction cell. All the experiments were conducted at $298 \pm 0.5$ K. The experiments were conducted under two RHs (72% and 82%) at 75 ppm $SO_2$ and 75 ppm $NO_2$.

During the reaction, particles were monitored *in-situ* via a glass window on the top of the reaction cell using a Micro-Raman spectrometer (LabRam HR800, HORIBA Jobin Yvon, Kyoto, Japan) to obtain microscopic images and Raman spectra. A 514-nm excitation laser was used, and back scattering Raman signals were detected. The details of the instrument are described elsewhere (Liu et al., 2008; Zhao et al., 2011). Because the particles were larger than the laser spot in this study (~1.5 μm), confocal Raman mapping was used to measure the spectra at different locations on a particle to obtain the chemical information of the entire particle. The mapping area was rectangular and was slightly larger than the particle, with mapping steps of $1 \times 1$ μm. Raman spectra in the range of 800–3,900 $cm^{-1}$ were acquired with an exposure time of 1 s for each mapping point. Raman spectra were analyzed using Labspec 5 software (HORIBA Jobin Yvon). Raman peaks were fitted to Gaussian–Lorentzian

functions to obtain peak positions and peak areas at different locations on the particle. The peak areas were then
added together to obtain the peak area for the entire particle.

Particles of $CaCO_3$ (98%, Sigma-Aldrich, USA), with average diameters of about 7–10 μm as specified by

the supplier, were used in the experiments. The $CaCO_3$ particles were rhombohedron crystals; X-ray diffraction
analysis indicated that they were calcite (Fig. S2). Individual particles were prepared by dripping a dilute $CaCO_3$
suspended solution onto Teflon-FEP film using a pipette and then drying the sample in an oven at 80ºC for 10 h.

The amount of $CaSO_4$ as a reaction product was quantified based on Raman peak areas and particle sizes.

The details of the method are described in our previous study (Zhao et al., 2017). Briefly, the amount of reaction
product $CaSO_4$ formed was determined as a function of time using Raman peak areas. Raman peak areas were
converted to the amount of compound formed using a calibration curve obtained from pure $CaSO_4$ particles of
different sizes, which were determined according to microscopic images. The reaction rate, i.e., the sulfate
production rate, was derived from the amount of sulfate formed as a function of time. The reactive uptake
coefficient of $SO_2$ for sulfate formation (γ) was further determined from the reaction rate and collision rate of $SO_2$
on the surface of a single particle.

$$\gamma = \frac{\frac{d\{SO_4^{2-}\}}{dt}}{Z} \; .$$           (1)

$$Z = \frac{1}{4}cA_s[SO_2],$$           (2)

$$c = \sqrt{\frac{8RT}{\pi M_{SO_2}}} \; ,$$           (3)

where R is the gas constant, T is temperature, $M_{SO_2}$ is the molecular weight of $SO_2$, c is the mean molecular
velocity of $SO_2$, $A_s$ is the surface area of an individual particle, and Z is the collision rate of $SO_2$ on the surface of
a particle. $\{SO_4^{2-}\}$ indicates the amount of sulfate in the particle phase in moles. The average reaction rate and
surface area of particles during the multiphase reaction period were used to derive the reactive uptake coefficient.
The period was chosen to start after the induction period when ~10% of the final sulfate was formed. $[SO_2]$
indicates the concentration of $SO_2$ in the gas phase.

The influence of gas phase diffusion on reactive uptake was evaluated using the resistor model described by

Davidovits et al. (2006) and references therein, as well as using the gas phase diffusion correction factor for a
reactive uptake coefficient according to the method described by Pöschl et al. (2007). The reactive uptake of $SO_2$
was found to not be limited by gas phase diffusion (see details in the Supplement S1).

In addition, we conducted experiments of the reaction $SO_2$ with only $O_2$ on both $CaCO_3$ solid particles and

internally mixed $CaCO_3/Ca(NO_3)_2$ particles (with $CaCO_3$ embedded in $Ca(NO_3)_2$ droplets), while keeping other
conditions the same as the reaction of $SO_2$ with an $O_2/NO_2$ mixture. These experiments of the multiphase
oxidation of $SO_2$ by $O_2$ can help determine the role of $NO_2$ in the reaction of $SO_2$ with an $O_2/NO_2$ mixture.
**3  Results and discussion**
**3.1  Reaction products and changes in particle morphology**

Figure 1 shows the Raman spectra of a $CaCO_3$ particle during the multiphase reaction of $SO_2$ with $O_2/NO_2/H_2O$

on its surface. The peak at 1,087 $cm^{-1}$ was assigned to the symmetric stretching of carbonate ($\nu_s(CO_3^{2-})$)

(Nakamoto, 1997). During the reaction, the peak at 1,087 $cm^{-1}$ decreased continuously and finally disappeared as new peaks were observed. The peak at 1,050 $cm^{-1}$ was assigned to the symmetric stretching of nitrate ($\nu_s(NO_3^-)$). The peaks at 1,010 $cm^{-1}$ and 1,136 $cm^{-1}$ were assigned to the symmetric stretching ($\nu_s(SO_4^{2-})$) and asymmetric stretching($\nu_{as}(SO_4^{2-})$) of sulfate in gypsum ($CaSO_4 \cdot 2H_2O$), respectively (Sarma et al., 1998). In addition, after the reaction, a broad envelope in the range of 2,800–3,800 $cm^{-1}$ assigned to the stretching of the OH bond in water molecules was observed. Above this envelope, there were two peaks at 3,408 $cm^{-1}$ and 3,497 $cm^{-1}$, which were assigned to OH bond stretching in crystallization water of $CaSO_4 \cdot 2H_2O$ (Sarma et al., 1998; Ma et al., 2013).

During the multiphase reaction with the $SO_2/O_2/NO_2/H_2O$ mixture, the $CaCO_3$ particles displayed a remarkable change in morphology. The original $CaCO_3$ particle was a rhombohedron crystal (Fig. 2, panel i, a). As the reaction proceeded, its edges became smoother and later a transparent droplet layer formed, which had a newly formed solid phase embed in it (Fig. 2, panel i, d). The size of the new solid phase grew during the reaction (Fig. 2, panel i, d–f) and it seemed to contain many micro-crystals. Raman mapping revealed that the new solid phase consisted of $CaSO_4 \cdot 2H_2O$ (Fig. 2, panel iv), and the surrounding aqueous layer consisted of $Ca(NO_3)_2$ (Fig. 2, panel iii).

The particle morphology change shown in Fig. 2 was significantly different from the morphology change in the direct reaction of $SO_2$ with $NO_2$ (Zhao et al., 2017), where the $CaCO_3$ particle was first converted to a spherical $Ca(NO_3)_2$ droplet and then needle-shaped $CaSO_4$ crystals formed inside the droplet (Zhao et al., 2017). Moreover, the amount of $CaSO_4$ formed in this study was much higher than that in the direct reaction of $SO_2$ with $NO_2$. The $CaSO_4$ solid particle constituted a significant fraction of the volume of the droplet, while in the direct reaction of $SO_2$ with $NO_2$ the few needle-shaped $CaSO_4$ crystals that formed only constituted a small fraction of the droplet volume (Zhao et al., 2017).

**3.2 Reaction process**

During the reaction, the amounts of carbonate, nitrate, and sulfate were determined as a function of time, as shown in Fig. 3. At the beginning of the reaction, the amount of carbonate decreased slowly, while the amount of nitrate and sulfate increased slowly. After a period of induction of around 50 min, the reaction accelerated significantly, leading to a rapid consumption of carbonate and production of nitrate and sulfate. The decrease in the amount of carbonate and the increase in the amount of nitrate was because carbonate reacted continuously with $NO_2$ and $H_2O$, forming $Ca(NO_3)_2$. The detailed mechanism of the multiphase reaction of carbonate with $NO_2$ and $H_2O$ were discussed in our previous studies (Li et al., 2010; Zhao et al., 2017). The mechanism of sulfate formation is discussed in detail in Section 3.4 of the present study. Finally, the carbonate was completely consumed, and the amounts of nitrate and sulfate levelled off.

Figure 3 shows that nitrate and sulfate were formed simultaneously during the reaction. This contrasts with the observations made during the direct reaction of $SO_2$ with $NO_2$, where nitrate was formed first, and sulfate was essentially formed after the complete conversion of $CaCO_3$ particles to $Ca(NO_3)_2$ droplets (Zhao et al., 2017). Moreover, the time taken for carbonate to be completely consumed was longer in this study than in the direct reaction of $SO_2$ with $NO_2$ (~120 vs. ~40 min) when other conditions were kept the same (Zhao et al., 2017).

## 3.3 Reactive uptake coefficient of $SO_2$

The reactive uptake coefficients of $SO_2$ for sulfate formation (γ) in the reaction of $SO_2$ with the $O_2/NO_2/H_2O/N_2$ mixture on $CaCO_3$ with various $O_2$ concentrations are shown in Table 1. The value of γ for the reaction of $SO_2$ with $O_2/NO_2$ at three $O_2$ concentrations (5, 20, and 86%) was in the range of $(0.35–1.7) \times 10^{-5}$, and was $1.2 \times 10^{-5}$ in synthetic air. This latter value was 2–3 orders of magnitude higher than that for the reaction of $SO_2$ directly with $NO_2$ under similar conditions (Zhao et al., 2017). When other conditions were kept constant, γ increased with the $O_2$ concentration. This indicates that $O_2$ played a key role in enhancing the oxidation rate of $SO_2$.

The role of $O_2$ in enhancing the reactive uptake of $SO_2$ reported here is consistent with the findings in some previous studies. For example, Littlejohn et al. (1993)'s data showed that sulfite oxidation rate increases with the $O_2$ concentration (0–5% by volume). Shen and Rochelle (1998) also found that in the presence of $O_2$, the aqueous sulfite oxidation rate is enhanced. By investigating the oxidation of $SO_2$ by $NO_2$ in monodispersed water droplets growing on carbon nuclei, Santachiara et al. (1990) found that sulfate formation rate with 2% $O_2$ is much higher than that without $O_2$. Yet, our findings, as well as those in the studies referred to above, are in contrast to those reported by Lee and Schwartz (1983), who found that changing from $N_2$ to air as a carrier gas only increases bisulfite oxidation rate by no more than 10%. The difference between our study and Lee and Schwartz (1983) could be due to the difference in $O_2$ diffusion from gas to the condensed phase and the different mechanisms between the multiphase reaction on particles and the aqueous reaction.

Only few studies have reported the S(IV) oxidation rate in the reaction of S(IV) with $O_2/NO_2$ mixtures (Turšič et al., 2001; Littlejohn et al., 1993). However, due to the limiting step by the aqueous phase mass transfer, it is difficult to quantitatively compare the reaction rates in those studies with the uptake coefficient in our study and the rate constants determined by Lee and Schwartz (1983) and Clifton et al. (1988). For example, a rate constant of $2.4 \times 10^3$ $mol^{-1}$ $L$ $s^{-1}$ (at pH 3) can be derived from the results of Turšič et al. (2001), which is much lower than the values reported by Lee and Schwartz (1983) and Clifton et al. (1988). This can be attributed to the limiting step by the aqueous-phase mass transfer because the characteristic mixing time in the aqueous phase in Turšič et al. (2001) was likely much longer than that of Lee and Schwartz (1983) (1.7–5.3 s), according to the $HSO_3^-$ concentration time series reported by Turšič et al. (2001).

It is important to note that the concentrations of $NO_2$ and $SO_2$ used in this study are much higher than those in the ambient atmosphere. High concentrations of reactant gases are often used in laboratory studies in order to simulate the ambient reactions at the time scale of days or weeks and to get high signal-to-noise ratios for detecting products within minutes or hours. In the ambient atmosphere, reactive uptake coefficient of $SO_2$ should be lower than that in this study due to the lower $NO_2$ concentrations when other conditions are comparable and the chemical/physical processes observed in this study, such as changes in particle composition, phase, hygroscopicity, and pH should be much slower due to the lower concentrations of $NO_2$ and $SO_2$.

## 3.4 Reaction mechanism

In the multiphase reaction of $SO_2$ with $O_2/NO_2/H_2O$ on $CaCO_3$ particles, we found that $CaCO_3$ reacted with $NO_2$ and $H_2O$ and produced $Ca(NO_3)_2$, which deliquesced, forming liquid water, and provided a site for the

aqueous oxidation of $SO_2$. This process is similar to the direct reaction of $SO_2$ with $NO_2$ on $CaCO_3$ particles. The
details of this part of the reaction mechanism were discussed in our previous study (Zhao et al., 2017).
Once the aqueous phase was formed, $SO_2$ could undergo multiphase reactions with $O_2/NO_2$. The mechanism
of the direct aqueous reaction of S(IV) with $NO_2$ in the absence of $O_2$ is complex. Previous studies have proposed
two different mechanisms for the reaction. One involves $SO_3^{\bullet-}$ radical formation (Littlejohn et al., 1993; Shen and
Rochelle, 1998; Turšič et al., 2001) (referred as "free-radical chain" mechanism, while the other involves the
formation of $NO_2$–S(IV) complexes (Clifton et al., 1988), but no radical formation (referred as "$NO_2$–S(IV)
complex" mechanism).
According to the $NO_2$–S(IV) complex mechanism, the presence of $O_2$ should not affect the $SO_2$ oxidation
rate; however, in this study, a substantial enhancement in the $SO_2$ oxidation rate was observed in the presence of
$O_2$ compared with that in the absence of $O_2$. Therefore, the $NO_2$–S(IV) complex mechanism was less likely to
have been important in this study.
In the free-radical chain mechanism, the $SO_3^{\bullet-}$ radical is proposed to be formed (R8, Table 2), which is based
on the observation of $S_2O_6^{2-}$ formation, with $S_2O_6^{2-}$ known to be the combination reaction product of $SO_3^{\bullet-}$
(Eriksen, 1974; Hayon et al., 1972; Deister and Warneck, 1990; Brandt et al., 1994; Waygood and McElroy,
1992). In addition to $SO_4^{2-}$ and $NO_2^{-}$, $S_2O_6^{2-}$ was detected with an appreciable yield using Raman spectroscopy,
following the reaction of $NO_2$ with aqueous sulfite (Littlejohn et al., 1993). $S_2O_6^{2-}$ was also observed in the
aqueous oxidation of bisulfite in an $N_2$-saturated solution in the presence of Fe(III) using ion-interaction
chromatography (Podkrajšek et al., 2002). The $SO_3^{\bullet-}$ radical can react via two pathways, forming either $S_2O_6^{2-}$ or
$SO_4^{2-}$ (R9–R11, Table 2). The reactions R9–R11 have been well established in studies of S(IV) oxidation by
other pathways, including OH oxidation, photo-oxidation, and transition metal catalyzed oxidation (Eriksen, 1974;
Hayon et al., 1972; Deister and Warneck, 1990; Brandt et al., 1994; Brandt and Vaneldik, 1995; Waygood and
McElroy, 1992). In addition, although previous studies have not reported the direct observation of the $SO_3^{\bullet-}$
radical in the aqueous reaction of S(IV) with $NO_2$, $SO_3^{\bullet-}$ was observed in the reaction of $NO_2^{-}$ with $SO_3^{2-}$ in an
acidic buffer solution (pH = 4.0) using electron spin resonance (ESR) (Shi, 1994). Because $NO_2^{-}$ is formed in the
aqueous reaction of $SO_2$ with $NO_2$, and $S_2O_6^{2-}$ as the combination reaction product of $SO_3^{\bullet-}$ is observed (Littlejohn
et al., 1993), $SO_3^{\bullet-}$ formation is plausible.
In the presence of $O_2$, the $SO_3^{\bullet-}$ radical can react rapidly with $O_2$, forming the $SO_5^{\bullet-}$ radical (R12, Table 2).
Following this reaction, a number of chain reactions can occur to ultimately form sulfate (Littlejohn et al., 1993;
Seinfeld and Pandis, 2006; Shen and Rochelle, 1998) (R13–R16, Table 2). Littlejohn et al. (1993) observed that
the amount of $S_2O_6^{2-}$ relative to $SO_4^{2-}$ formed in the aqueous reaction of $NO_2$ with sulfite decreases in the
presence of $O_2$ compared with the reaction in the absence of $O_2$. At low $NO_2$ concentrations (< 5 ppm), $S_2O_6^{2-}$ is
undetectable in the presence of $O_2$. This indicates that $O_2$ suppresses the reaction pathway of $S_2O_6^{2-}$ formation
(R9, Table 2). Because the $SO_3^{\bullet-}$ radical can react rapidly with $O_2$, forming the $SO_5^{\bullet-}$ radical, and would therefore
be consumed, the suppression of $S_2O_6^{2-}$ formation can be attributed to the reaction of $SO_3^{\bullet-}$ with $O_2$ (R12, Table
2). The reactions R12–R16 have been well established by studies of the oxidation of S(IV) by OH or
photo-oxidation, and all the radicals have been observed (Hayon et al., 1972; Huie et al., 1989; Huie and Neta,
1987; Chameides and Davis, 1982; Seinfeld and Pandis, 2006).
The free-radical chain mechanism is consistent with the findings of this study and is therefore more plausible.
The enhancement of the $SO_2$ oxidation rate in the reaction of $SO_2$ with $O_2/NO_2/H_2O$ on $CaCO_3$ particles
compared with that in the direct reaction of $SO_2$ with $NO_2/H_2O$ was attributed to $O_2$. Although during the reaction
in the absence of $O_2$—i.e., the direct oxidation of $SO_2$ by $NO_2$—the $SO_3^{\bullet-}$ radical can be formed (R7), the reaction
chain cannot propagate (R12–R16). Therefore, the S(IV) oxidation rate and the reactive uptake coefficient of $SO_2$
were much lower than that in the presence of $O_2$. According to the difference between the reactive uptake
coefficient in this study and in the direct reaction of $SO_2$ with $NO_2$ (Zhao et al., 2017), the sulfate production rate
via chain reactions due to the presence of $O_2$ (20%) was 2–3 orders of magnitude faster than the direct oxidation
of $SO_2$ by $NO_2$. This indicates that sulfate production in the reaction of $SO_2$ with $O_2/NO_2$ was largely due to $O_2$
oxidation via the chain reaction pathway, i.e., "autoxidation" of S(IV), rather than the direct oxidation of $SO_2$ by
$NO_2$ and thus $O_2$ was the main oxidant of $SO_2$.
In addition to the two mechanisms above, Spindler et al. (2003) proposed a reaction mechanism involving
first $NO_2$–S(IV) complex formation and subsequent $SO_3^{\bullet-}$ radical formation (R3, R7). $NO_2$–S(IV) complex may
establish an equilibrium with $SO_3^{\bullet-}$ in contrast to the direct formation of $SO_3^{\bullet-}$ via the reaction of $NO_2$ with $SO_2$.
Higher concentration of $O_2$ favors the conversion of $SO_3^{\bullet-}$ to $SO_5^{\bullet-}$ and thus can reduce the $SO_3^{\bullet-}$ concentration,
shifting the equilibrium to the product side and promoting the overall S(IV) oxidation. $O_2$ can act in a similar way
as in the free-radical chain mechanism. Admittedly, we cannot rule out the possibility $NO_2$–S(IV) complex
formation. But such a mechanism can still be classified as the free-radical chain mechanism since the S(IV)
oxidation still proceeds via the radical chain reactions.Although the direct oxidation of $SO_2$ by $NO_2$ only
accounted for a very small fraction of sulfate formation, $NO_2$ played an important role in the $SO_2$ oxidation by
initiating the chain reactions via the production of the $SO_3^{\bullet-}$ radical (R7). In the experiment without $NO_2$, but with
other reaction conditions the same, we were unable to detect sulfate after 5 h of reaction. This indicates that $O_2$ by
itself cannot initiate the chain reactions (although it favors chain propagation), and that the oxidation of $SO_2$ by
$O_2$ was slow. The effect on the $SO_2$ oxidation rate when both $NO_2$ and $O_2$ were present was much higher than the
sum of the effect of $NO_2$ and $O_2$. We refer to this effect as the synergy of $NO_2$ and $O_2$, which resulted in the fast
oxidation of $SO_2$ to form sulfate in this study. This effect is similar to a "ternary" reaction found with the reaction
of $NO_2$–particles–$H_2O$ or $SO_2$–particles–$O_3$ (Zhu et al., 2011), where the reaction rate can be much faster than the
sum of the reaction rates for the reaction of the second and third reactant with the first reactant. In addition to
acting as the initiator of chain reactions, $NO_2$ also contributed to the formation of the aqueous phase through the
reaction with $CaCO_3$, forming $Ca(NO_3)_2$ as discussed above, which provided a site for S(IV) oxidation.
Based on the discussion above, we summarize the reaction mechanism that occurred in this study in Table 2.
The reactions are classified as chain initiation, chain propagation, and chain termination. The dominant S(IV)
species depends on pH. Due to the fast dissociations of $SO_2\bullet H_2O$ and $HSO_3^-$, reactions consuming one of these
S(IV) species will result in instantaneous re-establishment of the equilibria between them (Seinfeld and Pandis,
2006). In this study, the pH of the aqueous layer of $Ca(NO_3)_2$ may change dynamically with time during the
reaction and may not be completely homogeneous within the aqueous droplet. The pH values could vary between
~3 and ~7.6. In the surface of the aqueous layer, pH was mainly determined by the gas–aqueous equilibrium of
$SO_2$, and was estimated to be ~3. In the vicinity of the $CaCO_3$ core, pH was mainly determined by the hydrolysis
of carbonate and was estimated to be ~7.6. It is likely that both $HSO_3^-$ and $SO_3^{2-}$ were present, and the dominant
species depended on the reaction time and location within the aqueous droplet. Nevertheless, to make the reaction
mechanism clearer, $HSO_3^-$ was used in the reaction equations. Similar reaction equations are also applicable to
$SO_3^{2-}$ because of the fast dissociations of $SO_2 \cdot H_2O$ and $HSO_3^-$. Overall, the reaction can be written as follows,
which clearly shows that $O_2$ was the main oxidant for sulfate formation:
$$2NO_2(aq) + 2HSO_3^-(aq) + (0/1)O_2 \rightarrow 2NO_2^-(aq) + S_2O_{6/8}^{2-}(aq) + 2H^+(aq), \qquad \text{(R19)}$$

$$nO_2 + 2nHSO_3^-(aq) \rightarrow 2nSO_4^{2-}(aq) + 2nH^+(aq), \qquad \text{(R20)}$$

where n >> 1. Once sulfuric acid was formed, it reacted with $CaCO_3$, forming $CaSO_4$:
$$CaCO_3(s) + SO_4^{2-}(aq) + 2H^+(aq) + H_2O(aq) \rightarrow CaSO_4 \cdot 2H_2O(s) + CO_2(g). \qquad \text{(R21)}$$

As mentioned above, compared with the direct reaction of $SO_2$ with $NO_2$, $CaCO_3$ was consumed more
slowly in the reaction with $O_2/NO_2$. There were two possible reasons for this. First, the $CaSO_4 \cdot 2H_2O$ formed in
the reaction could cover the $CaCO_3$ surface and partly suppress the diffusion of aqueous ions, such as protons,
and also limit the contact of reactants with the surface of the $CaCO_3$ particles, thus reducing the $CaCO_3$
consumption rate. Second, compared with the direct reaction of $SO_2$ with $NO_2$, a much higher fraction of $CaCO_3$
was converted to $CaSO_4 \cdot 2H_2O$ instead of $Ca(NO_3)_2$ due to the fast production of $CaSO_4 \cdot 2H_2O$. Therefore, the
volume of a $Ca(NO_3)_2$ droplet was much smaller than that in the direct reaction of $SO_2$ with $NO_2$ for a given
$CaCO_3$ particle. Because the uptake rate of $NO_2$ was proportional to the droplet surface area and the $NO_2$
hydrolysis rate was proportional to the droplet volume, the rate of nitric acid production from $NO_2$ hydrolysis and
its reaction rate with $CaCO_3$ were reduced. Therefore, the $CaCO_3$ particles were consumed more slowly in the
reaction with $O_2/NO_2$.
**4 Conclusion and implications**
We investigated the multiphase reaction of $SO_2$ with $O_2/NO_2/H_2O$ on $CaCO_3$ particles. The reaction
converted $CaCO_3$ particles to $Ca(NO_3)_2$ droplets, in which $CaSO_4 \cdot 2H_2O$ was embedded and accounted for a
significant fraction of the droplet volume by the end of the reaction. The $Ca(NO_3)_2$ droplet formed by the reaction
of $CaCO_3$ with $NO_2$ provided a site for the multiphase oxidation of $SO_2$. Generally, nitrate and sulfate were
formed simultaneously. The reactive uptake coefficient of $SO_2$ for sulfate formation in the reaction of $SO_2$ with
$NO_2/H_2O$ in synthetic air was determined to be around $10^{-5}$. Compared with the reaction of $SO_2$ with $NO_2$ on a
$CaCO_3$ particle in the absence of $O_2$, i.e., the direct oxidation of $SO_2$ by $NO_2$ in $N_2$, sulfate production rate in the
reaction of $SO_2$ with $O_2/NO_2$ was enhanced by 2–3 orders of magnitude. According to the findings of this study
and the existing literature, $SO_2$ oxidation likely proceeded via a free-radical chain reaction mechanism. $O_2$ was
the main oxidant of $SO_2$, and $NO_2$ mainly acted as an initiator of the chain reactions. The synergy of $NO_2$ and $O_2$
resulted in the fast oxidation of $SO_2$. The absence of either $NO_2$ or $O_2$ led to much slower $SO_2$ oxidation.
Using a method developed in our previous study (Zhao et al., 2017), we assessed the importance of the
multiphase oxidation of $SO_2$ by $O_2$ in the presence of $NO_2$ by estimating the lifetime of $SO_2$ due to multiphase
reactions and the lifetime due to the gas phase reaction (with the OH radical). The lifetime of $SO_2$ due to the
multiphase reaction of $SO_2$ with $O_2/NO_2$ was estimated to be around 20 days using the reactive uptake coefficient
of $SO_2$ ($1.2 \times 10^{-5}$) and the typical particle surface area concentration for mineral aerosols in winter in Beijing
($6.3 \times 10^{-6}$ cm$^2$ cm$^{-3}$) (Huang et al., 2015). This lifetime is comparable to the lifetime of $SO_2$ due to the gas phase

reaction with OH, which is ~12 days assuming that the daytime OH concentration is $1 \times 10^6$ molecules $cm^{-3}$ (Lelieveld et al., 2016; Prinn et al., 2005). Therefore, we conclude that the multiphase oxidation of $SO_2$ by $O_2$ in the presence of $NO_2$ is likely to be an important source of sulfate and a sink of $SO_2$ in the ambient atmosphere, and can play a significant role in the sulfate formation during severe haze episodes, such as those that frequently occur in China. During haze episodes, there are high concentrations of $SO_2$ and $NO_2$ and relative humidity is often high (Zhang et al., 2014; Wang et al., 2016; Zheng et al., 2015b). Under these conditions, the multiphase oxidation of $SO_2$ by $O_2$ in the presence of $NO_2$ could proceed rapidly, forming sulfate. The enhanced sulfate concentration due to multiphase reactions and resulting aerosol water content can further promote the multiphase oxidation of $SO_2$. The reaction thus proceeds in a self-accelerating way. Therefore, it can contribute significantly to sulfate formation during haze episodes, which could explain the discrepancies between the observed and modelled sulfate concentrations (Cheng et al., 2016; Gao et al., 2016; Wang et al., 2016; Zheng et al., 2015a).

In addition, elucidating the mechanism of the multiphase reaction of $SO_2$ with $O_2/NO_2/H_2O$ in the atmosphere is important for the other atmospheric implications of the reaction besides sulfate formation. According to the reaction mechanism, the direct oxidation of $SO_2$ by $NO_2$ forms sulfate and nitrite, with a stoichiometry of 1:1, and nitrite can further form HONO under acidic conditions. The HONO could then evaporate into the atmosphere, where it would be an important source of OH radical. If $NO_2$ were the main oxidant of $SO_2$ in the multiphase reaction, the reaction would form one HONO molecule for every sulfate formed. Thus, the oxidation of $SO_2$ by $NO_2$ can simultaneously be an important source of HONO and OH radical, and $SO_2$ oxidation would be strongly coupled with reactive nitrogen chemistry. However, according to the mechanism of this study, $NO_2$ only acted as an initiator of the chain reactions in $SO_2$ oxidation and essentially all the $SO_2$ was oxidized by $O_2$. Therefore, the amount of HONO formation per sulfate formed was trivial. The oxidation of $SO_2$ by $O_2/NO_2$ is expected to be neither an important source of HONO and OH in the atmosphere nor to have a significant influence on reactive nitrogen chemistry.

In this study, we investigated the reaction of $SO_2$ with $O_2$ in the presence of $NO_2$ at three $O_2$ concentrations. The influence of the $O_2$ concentration was shown to be significant. Future experimental results with more $O_2$ concentration levels would provide more insights into the reaction mechanism and process.

In addition, in the ambient atmosphere, the internal mixing of organics with S(IV) in particles may influence the S(IV) oxidation rate by $O_2$ in the presence of $NO_2$. When organics is abundant in particles, for example during haze episodes in China, it can react with and thus scavenge radical anion carriers such as $SO_5^{\bullet-}$ and $SO_4^{\bullet-}$ (Herrmann, 2003; Herrmann et al., 2015; Huie, 1995). Therefore, the presence of internally mixed organics can reduce the effectivity of the potential radical reaction chain and of S(IV) oxidation, which can undermine the importance of the oxidation by $O_2$ in the presence of $NO_2$ in the overall S(IV) oxidation.

**Acknowledgements**

This work was supported by Natural Science Foundation Committee of China (41421064, 21190051, 40490265, 91544000) and Ministry of Science and Technology (Grant No. 2002CB410802).

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

**Table 1.** Reactive uptake coefficient of $SO_2$ for sulfate formation at 82% RH and at different $O_2$
concentrations.

| $SO_2/NO_2/O_2$ concentration | $\gamma$ |
|---|---|
| 75 ppm/ 75 ppm/ 86 % | $1.7\times10^{-5}$ |
| 75 ppm/ 75 ppm/ 20 % | $1.2\times10^{-5}$ |
| 75 ppm/ 75 ppm/ 5 % | $3.5\times10^{-6}$ |


**Table 2.** Summary of the mechanism of the reaction S(IV) with $O_2/NO_2$

| Step | Reactions | |
| --- | --- | --- |
| Initiation | $NO_2(aq) + HSO_3^-(aq) \rightarrow NO_2^-(aq) + SO_3^{\bullet-}(aq) + H^+(aq)$ | (R8) |
| Propagation | $SO_3^{\bullet-}(aq) + O_2(aq) \rightarrow SO_5^{\bullet-}(aq)$ | (R12) |
| | $SO_5^{\bullet-}(aq) + HSO_3^-(aq) \rightarrow HSO_5^-(aq) + SO_3^{\bullet-}(aq)$ | (R13) |
| | $HSO_5^-(aq) + HSO_3^-(aq) \rightarrow 2SO_4^{2-}(aq) + 2H^+(aq)$ | (R14) |
| | $SO_5^{\bullet-}(aq) + HSO_3^-(aq) \rightarrow SO_4^{2-}(aq) + SO_4^{\bullet-}(aq) + H^+(aq)$ | (R15) |
| | $SO_4^{\bullet-}(aq) + HSO_3^-(aq) \rightarrow SO_4^{2-}(aq) + SO_3^{\bullet-}(aq) + H^+(aq)$ | (R16) |
| Termination | $SO_3^{\bullet-}(aq) + SO_3^{\bullet-}(aq) \rightarrow S_2O_6^{2-}(aq)$ | (R9) |
| | $SO_3^{\bullet-}(aq) + SO_3^{\bullet-}(aq) \rightarrow SO_3^{2-}(aq) + SO_3$ | (R10) |
| | $SO_3(aq) + H_2O \rightarrow SO_4^{2-}(aq) + 2H^+(aq)$ | (R11) |
| | $SO_4^{\bullet-}(aq) + SO_4^{\bullet-}(aq) \rightarrow S_2O_8^{2-}(aq)$ | (R17) |
| | $SO_5^{\bullet-}(aq) + SO_5^{\bullet-}(aq) \rightarrow S_2O_8^{2-}(aq) + O_2(aq)$ | (R18) |

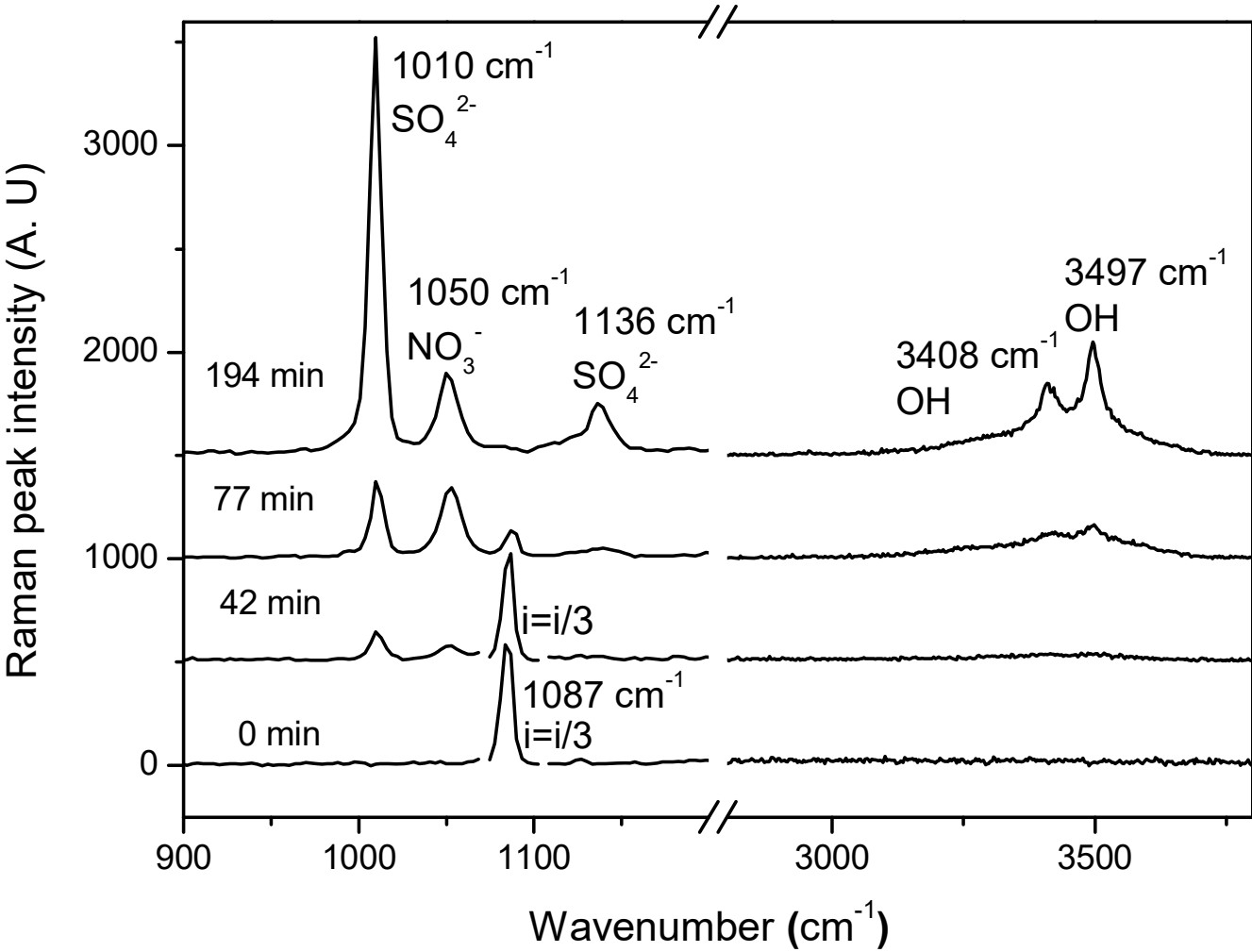


Figure 1. Raman spectra of a $CaCO_3$ particle during the multiphase reaction of $SO_2$ with $O_2/NO_2/H_2O$
on the particle. $SO_2$: 75 ppm, $NO_2$: 75 ppm, RH: 72%, $O_2$: 20%. The peak intensity of carbonate (1087
$cm^{-1}$) at 0 and 42 min was divided by three for clarity.

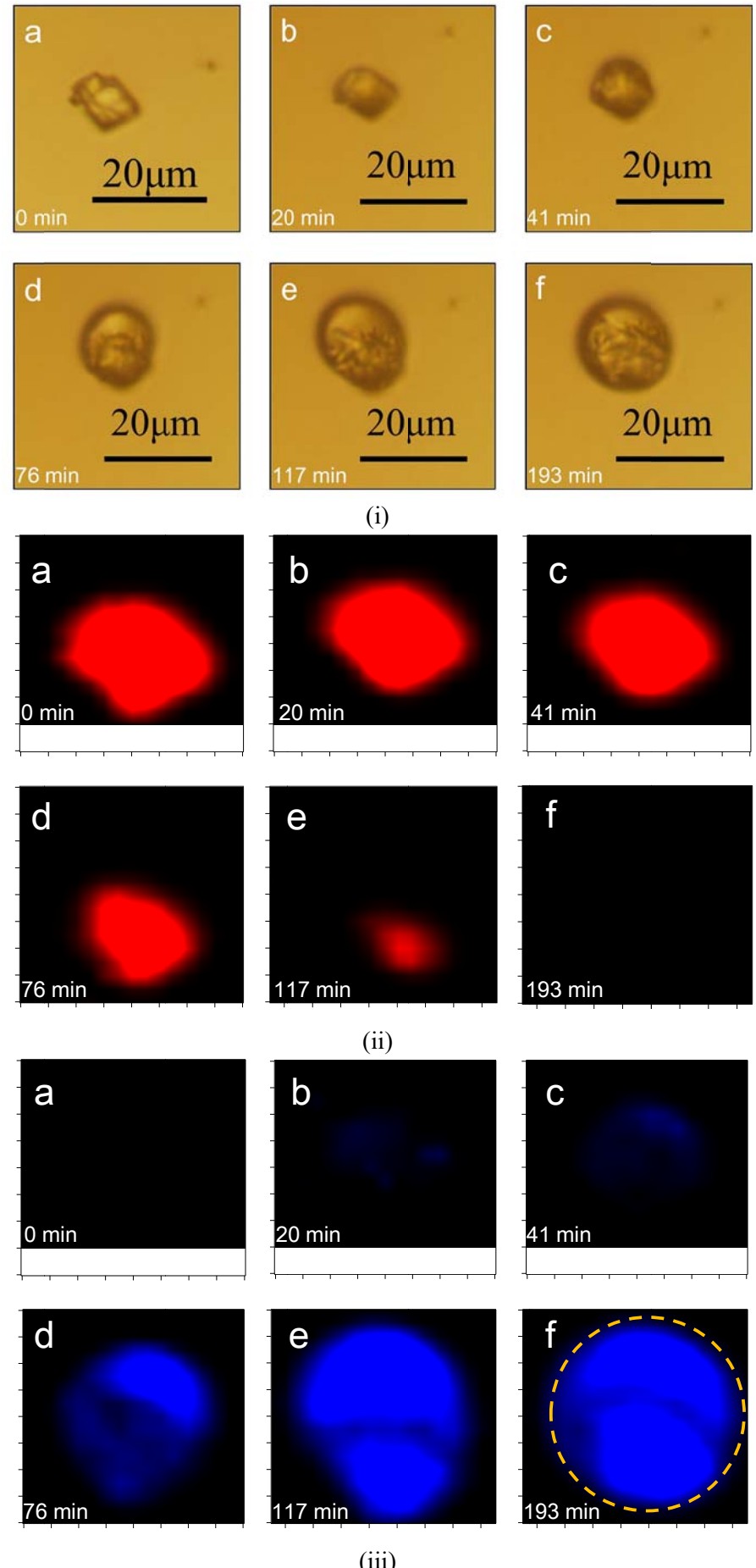


(i)


(ii)


(iii)

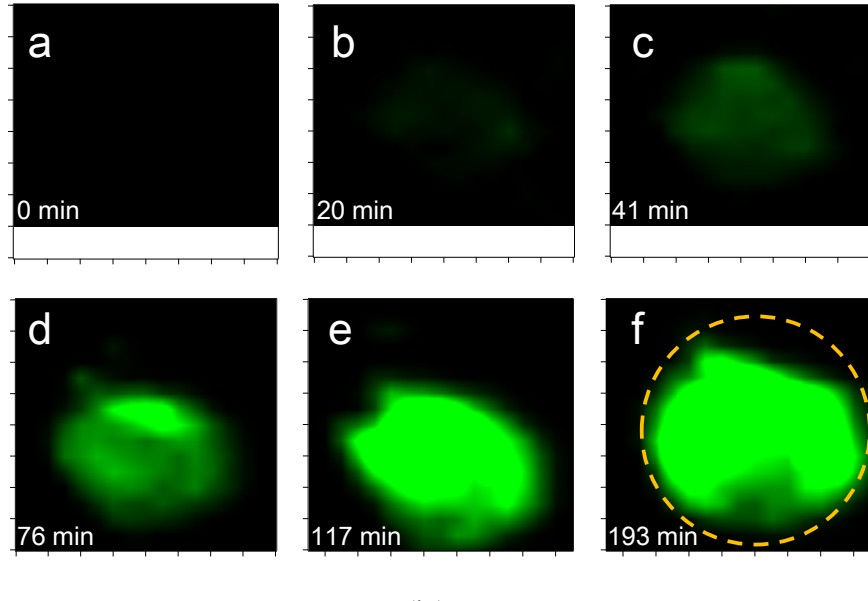



Figure 2. Microscopic image (i) and Raman mapping image of carbonate (ii), nitrate (iii), and sulfate (iv)
on the $CaCO_3$ particle during the multiphase reaction $SO_2$ with $O_2/NO_2/H_2O$ on the particle. A-f
corresponds to the reaction time of 0, 20, 41, 76, 117, and 193 min. $SO_2$: 75 ppm, $NO_2$: 75 ppm, RH:
72%, $O_2$: 20%. The mapping image of carbonate, nitrate, and sulfate are made using the peak area at
1050, 1010, and 1087 $cm^{-1}$, respectively. The red, blue, and green colors indicate the peak intensity of
carbonate, nitrate, and sulfate, respectively. The dashed lines in panel iii-f and iv-f indicate the shape of
the droplet at the end of the reaction.

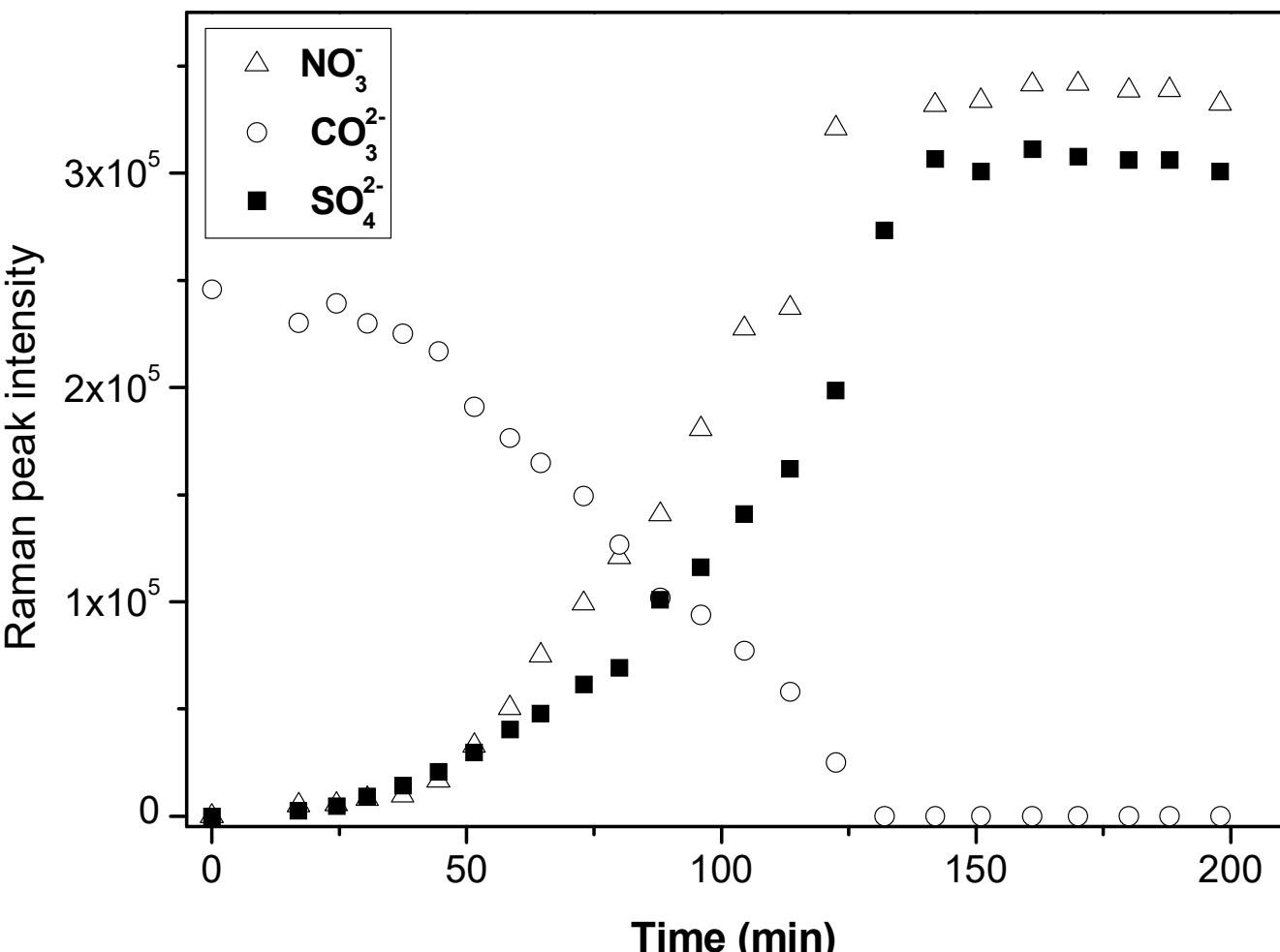


Figure 3. Time series of the Raman peak intensity of $NO_3^-$, $SO_4^{2-}$, and $CO_3^{2-}$ during the reaction of $SO_2$

with $O_2/NO_2/H_2O$ on $CaCO_3$ particles. $SO_2$: 75 ppm, $NO_2$: 75 ppm, RH: 72%, $O_2$: 20%. The intensity of

$NO_3^-$, $SO_4^{2-}$, and $CO_3^{2-}$ show the peak area at 1050, 1010, and 1087 cm$^{-1}$, respectively, in Raman spectra

obtained by Raman mapping.

519

520