# Peer review of "NO2-initiated multiphase oxidation of SO2 by O2 on CaCO3 particles"

_Atmospheric Chemistry and Physics, 2017_

## Referee Comment (RC1) · Anonymous Referee #1 · 27 Oct 2017

**General comments**

In this manuscript, the results on multiphase reaction of $SO_2$ on individual $CaCO_3$ particles in the presence of $NO_2$ and $O_2$ at RH 72% using Micro-Raman spectrometer with a flow reaction system are presented. The main conclusion is that $CaCO_3$ converts first to $Ca(NO_3)_2$ droplet (by the reaction with $NO_2$), where actually further aqueous $SO_2$ oxidation by $O_2$ takes place. The reactive uptake coefficient $\gamma$ of $SO_2$ determined on the basis of $SO_4^{2-}$ formation rate was ca. three orders of magnitude higher than that determined in the absence of $O_2$. On the basis of their results and mainly on literature data, they suggested that $NO_2$ first initiates a free-radical chain mechanism *via* reaction of $NO_2$ with $HSO_3^-/SO_3^{2-}$, where $SO_3^{\bullet-}$ radical is formed, which reacts quickly with $O_2$ to form $SO_5^{\bullet-}$, etc… The presented mechanism is well known and confirmed by many authors, and as expected the reaction under $O_2$ conditions is relatively fast and can be important source of sulfate.

As I have already pointed out in the previous review (Zhao et al., ACPD), I support the topic, mainly due still unresolved questions concerning high sulfate formation shown during heavily polluted episodes under haze conditions in China. And also, I like the approach used for studying processes on the micro level.

However, I found many mistakes (not only English language, but in general); the article is not well readable and many times confusing, many sentences are repeated with no need through the manuscript; thus, I cannot recommend it in the present form for publication in ACP.

Further, I again strongly suggest presenting the results for both systems $SO_2/NO_2/H_2O/N_2$ and $SO_2/NO_2/H_2O/O_2$ together in one article, although the authors of the first manuscript gave their reasons (in the responses) why to present separately. I think it is reasonable to show both together, first due to easier comparison, further due to easier discussion on differences in the mechanisms, etc. Anyway, the authors compare the results with the previous one during the whole manuscript. In addition, the experimental part is practically the same.

I highlight some of the main comments, questions and mistakes below. I will not expose the language mistakes, they are too many. Many parts of the manuscript should be rewritten.

**Specific comments**

Introduction: It needs to be rewritten.

1.  The authors should say something on well known and published mechanisms in the introduction. Discussion on the p.6/7 should partly be included here. Especially, the mechanism (R1-R6) does not fit on p.6, because it seems that is not important for their results.

2.    P. 2, line 39 and p. 8, line 243: Dissolution/absorption of $SO_2$ in aqueous solution results in total dissolved S(IV), i.e. three species $SO_2 x\ H_2O$, $HSO_3^-$ and $SO_3^{2-}$, which are in equilibrium; which species will prevail depends on pH! So, $H_2SO_3$ ($2H^+ + SO_3^{2-}$) are the same species as you have already written. In addition, it is not "rapid inter-conversion" between the species.

Experimental:

3.    RH is not mentioned in the experimental part, only in figure captions (RH 72%) and Table 1 (RH 82%). Is this fine or wrong?

4.    Can you say something on pH of the formed aqueous layer of $Ca(NO_3)_2$? If you know the pH you can say something more on mechanism; i.e. if it is above 6, than $SO_3^{2-}$ species are the main species which are involved.

Results and discussion:

5.    If you once define that you have a system $NO_2/H_2O/O_2$, where $O_2$ is from synthetic air, there is no need to repeat "in synthetic air" throughout the manuscript.

6.    Paragraph 3.2 An explanation on increasing concentration of $NO_3^-$ during the reaction is needed.

7.    P.5, line 146: The reactive uptake coefficient $\gamma$ of $SO_2$ was determined at three different $O_2$ concentrations, and not in the range 5-86%.

8.    P. 5, line 147: I can see that the increase in the reactive uptake coefficient $\gamma$ of $SO_2$ is ca. three orders (and not two to three) of magnitude higher than that determined in the absence of $O_2$.

9.    In the second paragraph of 3.3 you compare your results with the literature. Of course, that your results are different than that from Lee and Swartz (1983), due to many reasons, but probably the most important is their different approach. You can probably make some comparisons of your reaction rates with the rates got for the gas mixture $SO_2/NO_2$/air introduced into aqueous solution (Turšič et al., 2001).

10.   The mechanism shown on p.6 (from line 172 to 189) should be deleted here and just mentioned in the introduction. It is not important for the discussion, but can be written in one sentence why this mechanism is not possible.

11.   Check the reactions in the text and in the Table 2 (see R7-R8 in the text and those in the Table, R8–R10, R8a, R8b)! Anyway, it is no need to repeat; all important reactions in the Table are enough.
      If you know the pH, you can write the reaction with only one species, $HSO_3^-$ or $SO_3^{2-}$.

12. P.7, line 209-213: Nice study on $S_2O_6^{2-}$ species formation, although in a different system, can be found in Podkrajšek et al, Chemosphere 49 (2002). Whatever, the mechanism (and the reason) of its formation should be the same.

13. Better than "radical mechanism" is "a free-radical chain mechanism"

14. P. 8, Line 239: I do not agree that there is a synergy between $NO_2$ and $O_2$ (definition of synergy!); $NO_2$ only initiates the free-radical chain mechanism, and after the induction period, the reaction is relatively fast; and according to the proposed mechanism does not have other role, except in the first step when $Ca(NO_3)_2$ is formed in the reaction between $CaCO_3$ and $NO_2$. This part is now explained well in the first manuscript (Zhao et al., ACPD).

15. I also suggest excluding older references, and rather including only those after 1990.

16. The radicals throughout the manuscript are written incorrectly.

---

## Referee Comment (RC2) · Anonymous Referee #2 · 14 Nov 2017

I support the other referee's assessment, including the recommendation that Part 1 and Part 2 be combined into a single article. The articles don't stand on their own, and studying the reaction of SO2 with NO2 in the absence of O2, as in Part 1 - since O2 apparently plays a role in the reaction - is not relevant for atmospheric chemistry. I also agree that there are numerous English language errors in this manuscript.

The study and the results presented are interesting. However, I have doubts about the technical soundness of the approach. For one thing, it's impossible to understand the experimental approach based on what is written in section 2 of this manuscript. Yes, the reactor was a flow reactor, but where were the particles? Were they part of the flow? Or were gases flowing past the particles which are stationary on a surface? What were the particles like (size, shape, porosity, etc.)? How were they prepared and

dispersed? The manuscript is meaningless without these basic pieces of information. It is possible that they were mentioned in Part I of the manuscript, but I am being asked to review only this manuscript - and it must stand on its own at least to this extent.

I suspect that the particles were not part of the flow. In which case, did the authors consider the issue of gas phase diffusion limitations in their data analysis?

Was relative humidity actually measured or only inferred from mixing ratios of humid and dry air? RH is well know to be unpredictable in experiments, it should be measured directly.

Line 36: This may be a language issue but it is not appropriate to refer to a point of disagreement in the literature which has prompted detailed analysis and publications as "different opinions." Replace with "uncertainties in the pH value..." or something similar.

Line 56: Delete "O2 is abundant in the atmosphere," this is an atmospheric chemistry journal.

---

## Author Comment (AC1) · 28 Dec 2017

**Response to reviewer #1**

We thank the reviewer for carefully reviewing our manuscript; the comments are greatly appreciated. We will address all the comments and make major changes to the manuscript. In particular, we will substantially revise the section of "Introduction", "Experimental", and "Reaction mechanism". We believe that the revisions based on these comments will substantially improve our manuscript. In the following please find our responses to the comments one by one, and the corresponding changes to be made to the manuscript. The original comments are shown in italics. The changes to be made in the revised manuscript are highlighted.

*General comments*

*In this manuscript, the results on multiphase reaction of $SO_2$ on individual $CaCO_3$ particles in the presence of $NO_2$ and $O_2$ at RH 72% using Micro-Raman spectrometer with a flow reaction system are presented. The main conclusion is that $CaCO_3$ converts first to $Ca(NO_3)_2$ droplet (by the reaction with $NO_2$), where actually further aqueous $SO_2$ oxidation by $O_2$ takes place. The reactive uptake coefficient γ of $SO_2$ determined on the basis of $SO_4^{2-}$ formation rate was ca. three orders of magnitude higher than that determined in the absence of $O_2$. On the basis of their results and mainly on literature data, they suggested that NO2 first initiates a free-radical chain mechanism via reaction of $NO_2$ with $HSO_3^-/SO_3^{2-}$, where $SO_3^{\cdot-}$ radical is formed, which reacts quickly with $O_2$ to form $SO_5^{\cdot-}$, etc…The presented mechanism is well known and confirmed by many authors, and as expected the reaction under $O_2$ conditions is relatively fast and can be important source of sulfate.*

*As I have already pointed out in the previous review (Zhao et al., ACPD), I support the topic, mainly due still unresolved questions concerning high sulfate formation shown during heavily polluted episodes under haze conditions in China. And also, I like the approach used for studying processes on the micro level.*

*However, I found many mistakes (not only English language, but in general); the article is not well readable and many times confusing, many sentences are repeated with no need through the manuscript; thus, I cannot recommend it in the present form for publication in ACP.*

*Further, I again strongly suggest presenting the results for both systems $SO_2/NO_2/H_2O/N_2$ and $SO_2/NO_2/H_2O/O_2$ together in one article, although the authors of the first manuscript gave their reasons (in the responses) why to present separately. I think it is reasonable to show both together, first due to easier comparison, further due to easier discussion on differences in the mechanisms, etc. Anyway, the authors compare the results with the previous one during the whole manuscript. In addition, the experimental part is practically the same.*

*I highlight some of the main comments, questions and mistakes below. I will not expose the language mistakes, they are too many. Many parts of the manuscript should be rewritten.*

**Response:**

In the revised manuscript, we will make major changes to the manuscript in order to make the text more clear and improve readability. Moreover, we will have a thorough language check of the text of

this manuscript and correct the errors in language.

Regarding the arrangement of the two manuscripts, although they have some parts in common, the scientific questions under investigation and the chemistry in these two studies are substantially different. We have decided to arrange the two manuscripts as independent papers instead of companion papers to highlight their distinct features. The reasons for this arrangement are as follows:

1. The question whether the multiphase reaction of $SO_2$ directly with $NO_2$ is an important pathway for sulfate formation in the real atmosphere, e.g., during haze episodes in China, requires investigating the "pure" reaction of $SO_2$ with $NO_2$ without confounding effects of other oxidants. In another manuscript (Zhao et al., 2017), we address this question. And in this manuscript, we address a different question, i.e., whether the multiphase reaction of $SO_2$ with $O_2$ in the presence of $NO_2$ is an important reaction pathway of $SO_2$ oxidation; obviously, this reaction is more relevant to real atmospheric conditions.

2. We found that the multiphase reaction of $SO_2$ with $NO_2$ and the reaction of $SO_2$ with $O_2$ in the presence of $NO_2$ have significantly different chemistry, with different reaction mechanisms, products, and atmospheric implications.

i. The multiphase reaction of $SO_2$ directly with $NO_2$ involves two reactants whereas the reaction of $SO_2$ with $O_2/NO_2$ involves three reactants. In the former reaction $NO_2$ is the main oxidant of $SO_2$ while in the latter reaction $O_2$ is the main oxidant of $SO_2$ and $NO_2$ only acts as an initiator of chain reactions.

ii. According to the reaction mechanism, the main products in the multiphase reaction of $SO_2$ with $NO_2$ are sulfate and nitrite with a stoichiometry of 1:1. In contrast, the main product in the multiphase reaction of $SO_2$ with $O_2$ in the presence of $NO_2$ is sulfate and the ratio of sulfate to nitrite is much higher than 1:1 because nitrite is only formed in the chain initiation step.

iii. Due to the different reaction mechanism, the atmospheric implication of the reaction of $SO_2$ with $O_2$ in the presence of $NO_2$ is significantly different from the direct reaction of $SO_2$ with $NO_2$ because the former leads to much faster sulfate formation and has much less influence on reactive nitrogen chemistry in the atmosphere.

Based on these reasons, we will change the title of this manuscript to "$NO_2$-initiated Multiphase Oxidation of $SO_2$ by $O_2$ on $CaCO_3$ Particles" in the revised manuscript. Accordingly, in the revised manuscript we will delete the parts where we stated about the companionship of these two papers (lines 63-64, 80 in the manuscript). We will also adjust the wording in the abstract and introduction to reflect this change.

***Specific comments***

*Introduction**: It needs to be rewritten.*

*1. The authors should say something on well known and published mechanisms in the introduction. Discussion on the p.6/7 should partly be included here. Especially, the mechanism (R1-R6) does not fit on p.6, because it seems that is not important for their results.*

**Response:**

Agree. In the revised manuscript, we will make major changes to the introduction and add the published mechanism to the introduction.

As the reviewer suggested, in the introduction we will also include part of the discussion regarding the mechanism on the p.6-7. It will read:

"Despite such a reaction mechanism of $SO_2$ oxidation has been proposed, its role in the $SO_2$ oxidation in the ambient atmosphere is not well established. Most previous studies focus on the direct reaction of $SO_2$ with $NO_2$, including determining its rate constant (Lee and Schwartz, 1983; Clifton et al., 1988; Shen and Rochelle, 1998; Spindler et al., 2003; Nash, 1979; Huie and Neta, 1986). According to the reaction products and their yields (Lee and Schwartz, 1983; Clifton et al., 1988), the overall reaction equation of the direct reaction of $NO_2$ with $SO_2$ is as follows:

$$2NO_2(aq) + HSO_3^-(aq) + H_2O \rightarrow 2NO_2^-(aq) + SO_4^{2-}(aq) + 3H^+(aq), \qquad (R1)$$

$$2NO_2(aq) + SO_3^{2-}(aq) + H_2O \rightarrow 2NO_2^-(aq) + SO_4^{2-}(aq) + 2H^+(aq), \qquad (R2).$$

which were proposed to proceed via $NO_2$-S(IV) adduct complexes (Clifton et al., 1988).

$$NO_2(aq) + SO_3^{2-}(aq) \rightarrow [NO_2 - SO_3]^{2-}(aq). \qquad (R3)$$

$$NO_2(aq) + [NO_2 - SO_3]^{2-}(aq) \rightarrow [NO_2 - SO_3 - NO_2]^{2-}(aq). \qquad (R4)$$

$$[NO_2 - SO_3 - NO_2]^{2-}(aq) + OH^-(aq) \rightarrow [NO_2 - SO_4H - NO_2]^{3-}(aq). \qquad (R5)$$

$$[NO_2 - SO_4H - NO_2]^{3-}(aq) \rightarrow 2NO_2^-(aq) + SO_4^{2-}(aq) + H^+(aq). \qquad (R6)"$$

*2. P. 2, line 39 and p. 8, line 243: Dissolution/absorption of $SO_2$ in aqueous solution results in total dissolved S(IV), i.e. three species $SO_2x\ H_2O$, $HSO_3^-$ and $SO_3^{2-}$, which are in equilibrium; which species will prevail depends on pH! So, $H_2SO_3$ ($2H^+ + SO_3^{2-}$) are the same species as you have already written. In addition, it is not "rapid interconversion" between the species.*

**Response:**

Agree. In the revised manuscript, we will change "$H_2SO_3$" to "$SO_2 \cdot H_2O$". And we will change "rapid interconversion" to "fast dissociations of $SO_2 \cdot H_2O$ and $HSO_3^-$".

*Experimental:*

*3. RH is not mentioned in the experimental part, only in figure captions (RH 72%) and Table 1 (RH 82%). Is this fine or wrong?*

**Response:**

The values for RH in figures and Table 1 are correct. The experiments were carried out at two RH, 72% and 82%. The experiments with varied concentrations of $O_2$ were done at 82 % RH. For the experiments at 72% RH, the reactive uptake coefficients were not quantified. Therefore, only the reactive uptake coefficients at 82% RH are shown in Table 1. In the revised manuscript, we will elaborate the experimental part and clarified this point as follows.

"The experiments of this study were conducted under two RH (72% and 82%) at 75 ppm $SO_2$ and 75

ppm $NO_2$."

*4. Can you say something on pH of the formed aqueous layer of Ca(NO₃)₂? If you know the pH you can say something more on mechanism; i.e. if it is above 6, than SO₃²⁻ species are the main species which are involved.*

**Response:**

The pH of the aqueous layer of $Ca(NO_3)_2$ may not be completely homogeneous within the aqueous layer and may change dynamically with time during the reaction. In the surface of the aqueous layer pH was supposed to be lower, which was mainly determined by the gas-aqueous equilibrium of $SO_2$, and estimated to be ~3. In the vicinity of the $CaCO_3$ core, pH was supposed to be higher due to carbonate hydrolysis, and was estimated to be ~7.6. Additionally, in the beginning of the reaction the overall pH of the aqueous layer should be higher due to the larger $CaCO_3$ core and thinner aqueous layer while in the end of the reaction overall pH should be lower, which was mainly determined by the gas-aqueous equilibrium of $SO_2$. Therefore, it is likely that both $HSO_3^-$ and $SO_3^{2-}$ were present, and the dominant species depended on the reaction time and location within the aqueous droplet. In the revised manuscript, we will add short discussion on this point.

"The dominant S(IV) species depends on pH. Due to the fast dissociations of $SO_2 \cdot H_2O$ and $HSO_3^-$, reactions consuming one of these S(IV) species will result in instantaneous re-establishment of the equilibria between them (Seinfeld and Pandis, 2006). In this study, the pH of the aqueous layer of $Ca(NO_3)_2$ may change dynamically with time during the reaction and may not completely homogeneous within the aqueous droplet. The pH values could vary between ~3 and ~7.6. In the surface of the aqueous layer pH was mainly determined by the gas-aqueous equilibrium of $SO_2$, and estimated to be ~3. In the vicinity of the $CaCO_3$ core, pH was mainly determined by the hydrolysis of carbonate and estimated to be ~7.6. It is likely that both $HSO_3^-$ and $SO_3^{2-}$ were present, and the dominant species depended on the reaction time and location within the aqueous droplet. Nevertheless, in order to make the reaction mechanism more clear, $HSO_3^-$ is used in the reaction equations. Similar reaction equations are also applicable to $SO_3^{2-}$ because of the fast dissociations of $SO_2 \cdot H_2O$ and $HSO_3^-$."

*Results and discussion:*

*5. If you once define that you have a system NO₂/H₂O/O₂, where O₂ is from synthetic air, there is no need to repeat "in synthetic air" throughout the manuscript.*

**Response:**

Agree. In the revised manuscript, we will not use "in synthetic air" throughout the manuscript where it is not necessary.

*6. Paragraph 3.2 An explanation on increasing concentration of NO₃⁻ during the reaction is needed.*

**Response:**

Agree. The concentration of $NO_3^-$ increased during the reaction because $NO_3^-$ was continuously formed by the reaction of $CaCO_3$ with $NO_2$ and $H_2O$. In the revised manuscript, we will briefly

discuss this point.

"The decrease of carbonate and the increase of nitrate is because carbonate continuously reacted with $NO_2$ and $H_2O$ forming $Ca(NO_3)_2$. The detailed mechanism of the multiphase reaction of carbonate with $NO_2$ and $H_2O$ are discussed in our previous studies (Li et al., 2010; Zhao et al., 2017)."

*7. P.5, line 146: The reactive uptake coefficient γ of SO$_2$ was determined at three different O$_2$ concentrations, and not in the range 5-86%.*

**Response:**

Agree. In the revised manuscript, we will change "in presence of $O_2$ (5%-86%)" to "in the presence of $O_2$ with three concentrations (5%, 20%, and 86%)".

*8. P. 5, line 147: I can see that the increase in the reactive uptake coefficient γ of SO$_2$ is ca. three orders (and not two to three) of magnitude higher than that determined in the absence of O$_2$.*

**Response:**

The reactive uptake coefficient γ of $SO_2$ for the reaction with $O_2/NO_2$ in synthetic air ($1.2 \times 10^{-5}$) was around 370 times higher than that determined for the direct oxidation of $SO_2$ by $NO_2$ ($3.22 \times 10^{-8}$). Therefore, we described this difference as "two to three orders of magnitude".

*9. In the second paragraph of 3.3 you compare your results with the literature. Of course, that your results are different than that from Lee and Swartz (1983), due to many reasons, but probably the most important is their different approach. You can probably make some comparisons of your reaction rates with the rates got for the gas mixture SO$_2$/NO$_2$/air introduced into aqueous solution (Turšič et al., 2001).*

**Response:**

We thank the reviewer for the suggestion. Regarding the experimental approach, Turšič et al. (2001) studied the absorption of the $SO_2/NO_2$/air mixture into aqueous solution, which is indeed more similar to our study. However, due to the mass transfer limitations, it is difficult to directly compare the reaction rates in that study with ours and the rate constants in other studies. A rate constant of $2.4 \times 10^3$ $mol^{-1}$ L $s^{-1}$ (at pH 3) can be derived from the study of Turšič et al. (2001) , which is much lower than the values from the study of Lee and Schwartz (1983) and of Clifton et al. (1988). This is likely attributed to the limiting step by the aqueous phase mass transfer. The characteristic mixing time in the aqueous phase in the study of Turšič et al. (2001) is likely much longer than that of Lee and Schwartz (1983) (1.7-5.3 s) according to the time series of $HSO_3^-$ concentration in the study of Turšič et al. (2001), although it was not explicitly reported. Nevertheless, in the revised manuscript, we will add this comparison as follows.

"Only few studies have reported the S(IV) oxidation rate in the reaction of S(IV) and $O_2/NO_2$ mixtures (Turšič et al., 2001; Littlejohn et al., 1993). However, due to the limiting step by the aqueous phase mass transfer, it is difficult to quantitatively compare the reaction rates in those studies to the uptake coefficient in our study and the rate constants determined by Lee and Schwartz (1983) and

Clifton et al. (1988). For example, a rate constant of $2.4 \times 10^3$ mol$^{-1}$ L s$^{-1}$ (at pH 3) can be derived from the study of Turšič et al. (2001), which is much lower than the values from the study of Lee and Schwartz (1983) and of Clifton et al. (1988). This is likely attributed to the limiting step by the aqueous phase mass transfer because the characteristic mixing time in the aqueous phase in the study of Turšič et al. (2001) is likely much longer than that of Lee and Schwartz (1983) (1.7-5.3 s) according to the time series of HSO$_3^-$ concentration in the study of Turšič et al. (2001)."

*10. The mechanism shown on p.6 (from line 172 to 189) should be deleted here and just mentioned in the introduction. It is not important for the discussion, but can be written in one sentence why this mechanism is not possible.*

**Response:**

Agree. In the revised manuscript, we will delete the mechanism from line 172 to 189 and only mention them in the introduction, as the reviewer suggested. Instead, we will briefly discuss why this mechanism is not possible as follows.

"According to the NO$_2$-S(IV) adduct mechanism, the presence of O$_2$ should not affect the SO$_2$ oxidation rate, which is opposite to the substantial enhancement in the SO$_2$ oxidation rate observed in the presence of O$_2$ compared to that in the absence of O$_2$. Therefore, NO$_2$-S(IV) adduct mechanism is likely not important in this study."

*11. Check the reactions in the text and in the Table 2 (see R7-R8 in the text and those in the Table, R8–R10, R8a, R8b)! Anyway, it is no need to repeat; all important reactions in the Table are enough.*

*If you know the pH, you can write the reaction with only one species, HSO$_3^-$ or SO$_3^{2-}$.*

**Response:**

Agree. In the revised manuscript, we will change R8 to R7. We will delete R7-17 in the text and only show them in the table. Moreover, as discussed above (the response to comment 4) we will only show the reaction equations for HSO$_3^-$ for clarity, although these reaction equations are also applicable to SO$_3^{2-}$ because of the fast dissociations of SO$_2$•H$_2$O and HSO$_3^-$.

*12. P.7, line 209-213: Nice study on S$_2$O$_6^{2-}$ species formation, although in a different system, can be found in Podkrajšek et al, Chemosphere 49 (2002). Whatever, the mechanism (and the reason) of its formation should be the same.*

**Response:**

We thank the reviewer for raising this study on S$_2$O$_6^{2-}$ formation. In the revised manuscript, we will add this paper in our citation and briefly discussed it as follows.

"S$_2$O$_6^{2-}$ was detected with an appreciable yield besides SO$_4^{2-}$ and NO$_2^-$ using Raman spectroscopy in the reaction of NO$_2$ with aqueous sulfite (Littlejohn et al., 1993). It was also observed in the aqueous oxidation of bisulfite in N$_2$-saturated solution in the presence of Fe(III) using ion-interaction chromatography (Podkrajšek et al., 2002)."

*13. Better than "radical mechanism" is "a free-radical chain mechanism".*

**Response:**

Agree. In the revised manuscript, we will change the "radical mechanism" to "a free-radical chain mechanism".

*14. P. 8, Line 239: I do not agree that there is a synergy between $NO_2$ and $O_2$ (definition of synergy!); $NO_2$ only initiates the free-radical chain mechanism, and after the induction period, the reaction is relatively fast; and according to the proposed mechanism does not have other role, except in the first step when $Ca(NO_3)_2$ is formed in the reaction between $CaCO_3$ and $NO_2$. This part is now explained well in the first manuscript (Zhao et al., ACPD).*

**Response:**

By synergy, we meant that the overall effect on the $SO_2$ oxidation rate when both $NO_2$ and $O_2$ were present was higher than the sum of the effect of $NO_2$ and of $O_2$, although the reactions of $SO_2$ with $O_2$ and with $NO_2$ were not always simultaneous. As we have shown, the $SO_2$ oxidation rates in the direct reaction of $SO_2$ with $NO_2$ and with $O_2$ were both very low.

In the revised manuscript, we will further elaborate this discussion as follows.

"In the experiment without $NO_2$ while keeping other reaction conditions the same, we found that no sulfate was formed after 5 h of reaction. This indicates that $O_2$ by itself cannot intialize the chain reaction, although it favors chain propagation, and the oxidation of $SO_2$ by $O_2$ was slow. The effect on the $SO_2$ oxidation rate when both $NO_2$ and $O_2$ were present was much higher than the sum of the effect of $NO_2$ and of $O_2$. We refer to this effect as the synergy of $NO_2$ and $O_2$, which resulted in the fast oxidation of $SO_2$ forming sulfate in this study. Without either $NO_2$ or $O_2$, the reaction proceeded much slower. "

*15. I also suggest excluding older references, and rather including only those after 1990.*

**Response:**

We thank the reviewer for the suggestion. However, some of the references earlier than 1990 are also important for the discussion of this study such as Lee and Schwartz (1983). Since ACP does not limit the number of references, we think it may be more reasonable to keep these references.

*16. The radicals throughout the manuscript are written incorrectly.*

**Response:**

Agree. In the revised manuscript, we will check and correct the writing of radicals throughout the manuscript wherever there were mistakes.

**References**

[revised manuscript text omitted]

---

## Author Comment (AC2) · 28 Dec 2017

**Response to reviewer #2**

We thank the reviewer for carefully reviewing our manuscript and providing helpful comments. All the comments will be addressed in the revised manuscript and we believe that the revisions based on these comments will substantially improve our manuscript. In the following please find our responses to the comments one by one, and the corresponding changes to be made to the manuscript. The original comments are shown in italics. The changes to be made in the revised manuscript are highlighted.

*I support the other referee's assessment, including the recommendation that Part 1 and Part 2 be combined into a single article. The articles don't stand on their own, and studying the reaction of $SO_2$ with $NO_2$ in the absence of $O_2$, as in Part 1 - since $O_2$ apparently plays a role in the reaction - is not relevant for atmospheric chemistry. I also agree that there are numerous English language errors in this manuscript.*

**Response:**

We appreciate the reviewer's opinion on how to better present our studies. However, these two studies are substantially different regarding the scientific questions under investigation and the chemistry involved, although they have some links. We have decided to arrange the two manuscripts as independent papers instead of companion papers to highlight their distinct features. The reasons for this arrangement are as follows:

1. The question whether the multiphase reaction of $SO_2$ directly with $NO_2$ is an important pathway for sulfate formation in the real atmosphere, e.g., during haze episodes in China, requires investigating the "pure" reaction of $SO_2$ with $NO_2$ without confounding effects of other oxidants. In another manuscript (Zhao et al., 2017), we address this question. And in this manuscript, we address a different question, i.e., whether the multiphase reaction of $SO_2$ with $O_2$ in the presence of $NO_2$ is an important reaction pathway of $SO_2$ oxidation. Both reactions have their own relevance to the atmosphere.

2. We found that the multiphase reaction of $SO_2$ with $NO_2$ and the reaction of $SO_2$ with $O_2$ in the presence of $NO_2$ have significantly different chemistry, with different reaction mechanisms, products, and atmospheric implications.

i. The multiphase reaction of $SO_2$ directly with $NO_2$ involves two reactants whereas the reaction of $SO_2$ with $O_2/NO_2$ involves three reactants. In the former reaction $NO_2$ is the main oxidant of $SO_2$ while in the latter reaction $O_2$ is the main oxidant of $SO_2$ and $NO_2$ only acts as an initiator of chain reactions.

ii. According to the reaction mechanism, the main products in the multiphase reaction of $SO_2$ with $NO_2$ are sulfate and nitrite with a stoichiometry of 1:1. In contrast, the main product in the multiphase reaction of $SO_2$ with $O_2$ in the presence of $NO_2$ is sulfate and the ratio of sulfate to nitrite is much

higher than 1:1 because nitrite is only formed in the chain initiation step.

iii. Due to the different reaction mechanism, the atmospheric implication of the reaction of $SO_2$ with $O_2$ in the presence of $NO_2$ is significantly different from the direct reaction of $SO_2$ with $NO_2$ because the former leads to much faster sulfate formation and has much less influence on reactive nitrogen chemistry in the atmosphere.

Based on these reasons, we will change the title of this manuscript to "$NO_2$-initiated Multiphase Oxidation of $SO_2$ by $O_2$ on $CaCO_3$ Particles" in the revised manuscript. Accordingly, in the revised manuscript we will delete the parts where we stated about the companionship of these two papers (lines 63-64, 80 in the manuscript). We will also adjust the wording in the abstract and introduction to reflect this change.

Regarding the language errors, we will have a thorough language check of the text of this manuscript and correct the errors in language.

*The study and the results presented are interesting. However, I have doubts about the technical soundness of the approach. For one thing, it's impossible to understand the experimental approach based on what is written in section 2 of this manuscript. Yes, the reactor was a flow reactor, but where were the particles? Were they part of the flow? Or were gases flowing past the particles which are stationary on a surface? What were the particles like (size, shape, porosity, etc.)? How were they prepared and dispersed? The manuscript is meaningless without these basic pieces of information. It is possible that they were mentioned in Part I of the manuscript, but I am being asked to review only this manuscript - and it must stand on its own at least to this extent.*

**Response:**

Agree. In the revised manuscript, we will elaborate the experimental part. The details of experiments including the details that the reviewer is concerned with will be added and explicitly described. The revision will also reflect the change from the companion paper of our last manuscript to a completely independent paper. The revised experimental part will be as follows.

"The experiments were conducted using a flow reaction system and the setup is illustrated in Fig. S1. The experimental setup and procedure used have been described in details in previous studies (Zhao et al., 2017; Zhao et al., 2011; Liu et al., 2008). A gas mixture of $NO_2$, $SO_2$, $O_2$, $N_2$, and water vapor reacted with particles deposited on a substrate in the flow reaction cell. The concentrations of $SO_2$ and $NO_2$ were controlled using mass flow controllers by varying the flow rates of $SO_2$ (2000 ppm in high purity $N_2$, National Institute of Metrology P.R. China), $NO_2$ (1000 ppm in high purity $N_2$, Messer, Germany), and synthetic air [20% $O_2$ (high purity grade: 99.999%, Beijing Haikeyuanchang Practical Gas Co., Ltd.), 80% $N_2$ (high purity grade: 99.999%, Beijing Haikeyuanchang Practical Gas Co., Ltd.)]. Relative humidity was controlled by regulating the flow rates of reactant gases, dry synthetic air, and humidified synthetic air. Humidified synthetic air was prepared by bubbling synthetic air

through fritted glass in water. In addition, in some experiments $O_2$ concentrations were varied by regulating the mixing ratios of $O_2$ and $N_2$ in order to investigate the effect $O_2$. $SO_2/NO_2/O_2/H_2O$ mixtures flew through the reaction cell and reacted with individual stationary $CaCO_3$ particles deposited on a Teflon-FEP film substrate annealed to a silicon wafer. RH and temperature were measured using a hygrometer (HMT100, Vaisala) at the exit for gases of the reaction cell. Additionally, temperature was measured using another small temperature sensor (Pt 100, 1/3 DIN B; Heraeus) in the reaction cell. All the experiments were conducted at 298±0.5 K. The experiments of this study were conducted under two RH (72% and 82%) at 75 ppm $SO_2$ and 75 ppm $NO_2$.

During the reaction, particles were *in-situ* monitored via a glass window on the top of the reaction cell using a Micro-Raman spectrometer (LabRam HR800, HORIBA Jobin Yvon) to obtain microscopic images and Raman spectra. A 514 nm excitation laser was used and back scattering Raman signals were detected. The details of the instrument are described in previous studies (Liu et al., 2008; Zhao et al., 2011). Because particles are larger than the laser spot in this study (~1.5 μm), confocal Raman mapping was used to obtain the spectra on different points of a particle in order to get the chemical information of the entire particle. The mapping area is a rectangular slightly larger than the particle and mapping steps are 1×1 μm. Raman spectra in the range 800-3900 cm$^{-1}$ were acquired with exposure time of 1 s for each mapping point. Raman spectra were analyzed using Labspec 5 software (HORIBA Jobin Yvon). Raman peaks were fit to Gaussian-Lorentzian functions to obtain peak positions and peak areas on different points of the particle. The peak areas were then added up to get the peak area for the entire particle.

$CaCO_3$ (98%, Sigma-Aldrich, USA) with diameters about 7-10 μm on average as specified by the manufacture was used. $CaCO_3$ particles are rhombohedron crystals. X-ray diffraction analysis shows that $CaCO_3$ particles are calcite (Fig. S2). Individual particles were prepared by dripping dilute $CaCO_3$ suspended solution onto the Teflon-FEP film using a pipette and then drying the sample in the oven at 80 ºC for 10 h.

The amount of the reaction product $CaSO_4$ was quantified based on Raman peak areas and particle sizes. The details of the method are described in our previous study (Zhao et al., 2017). Briefly, the amount of reaction product $CaSO_4$ formed was followed as a function of time using Raman peak areas. Raman peak areas were converted to the amount of compound using a calibration curve obtained from pure $CaSO_4$ particles of different sizes, which were determined according to microscopic image. The reaction rate, that is, sulfate production rate, was derived from the amount of sulfate as a function of time. The reactive uptake coefficient of $SO_2$ for sulfate formation (γ) was further determined from the reaction rate and collision rate of $SO_2$ on surface of a single particle.

$$\gamma = \frac{\frac{d\{SO_4^{2-}\}}{dt}}{Z} \ . \tag{1}$$

$$Z = \frac{1}{4}cA_s[SO_2], \tag{2}$$

$$c = \sqrt{\frac{8RT}{\pi M_{SO_2}}} \quad , \tag{3}$$

where R is the gas constant, T is temperature, $M_{SO_2}$ is the molecular weight of $SO_2$, and c is the mean molecular velocity of $SO_2$, $A_s$ is the surface area of an individual particle, and Z is the collision rate of $SO_2$ on surface of a particle. $\{SO_4^{2-}\}$ indicates the amount of sulfate on the particle phase in mole. The average reaction rate and surface area of particles during the multiphase reaction period were used to derive the reactive uptake coefficient. The period was chosen to start after the induction period when ~10 % of final sulfate was formed. [$SO_2$] indicates the concentration of $SO_2$ in the gas phase.

In addition, we carried out experiments without $NO_2$, on either $CaCO_3$ solid particle, or $CaCO_3/Ca(NO_3)_2$ internally mixed particle with $CaCO_3$ embedded in $Ca(NO_3)_2$ droplet in order to investigate the multiphase oxidation of $SO_2$ by $O_2$ and thus elucidate the role of $NO_2$ in the reaction of $SO_2$ with $O_2/NO_2$ mixture."

*I suspect that the particles were not part of the flow. In which case, did the authors consider the issue of gas phase diffusion limitations in their data analysis?*

**Response:**

Particles were deposited on a Teflon-FEP film in this study.

We evaluated the influence of gas phase diffusion on the reactive uptake coefficient using the resistor model according to the study of Davidovits et al. (2006) and references therein (see details in the Supplement S1 below, to be added in the revised manuscript). The contribution of the resistance ($1/\Gamma_{diff}$) due to gas phase diffusion to the reactive uptake coefficient in this study was less than 0.1%. Therefore we conclude that the reactive uptake of $SO_2$ was not limited by gas phase diffusion. The same conclusion can also be drawn by calculating the gas phase diffusion correction factor for a reactive uptake coefficient according to the method in Pöschl et al. (2007) (Equation 20 in their study).

In the revised manuscript, we will briefly discuss this point.

"The influence of gas phase diffusion on reactive uptake was evaluated using the resistor model according to the study of Davidovits et al. (2006) and references therein as well as using the gas phase diffusion correction factor for a reactive uptake coefficient according to the method in Pöschl et al. (2007). The reactive uptake of $SO_2$ was found to be not limited by gas phase diffusion (see details in the Supplement S1)."

The Supplement S1 to be added in the revised manuscript is as follows.

"**S1. Influence of gas phase diffusion on reactive uptake coefficients**

The Influence of the gas phase diffusion on reactive uptake coefficients was analyzed using the resistor model according to the study of Davidovits et al. (2006) and the references therein.

$$\frac{1}{\gamma} = \frac{1}{\Gamma_{diff}} + \frac{1}{\alpha} + \frac{1}{\Gamma_{sat}+\Gamma_{rxn}} \tag{1}$$

where $\Gamma_{diff}$ is the transport coefficient in the gas phase, $1/\Gamma_{diff}$ is the resistance due to the diffusion in the gas phase. Similarly, $1/\Gamma_{sat}$ and $1/\Gamma_{rxn}$ are the resistance due to liquid phase saturation and liquid phase reaction, respectively. $\alpha$ is the mass accommodation coefficient of $SO_2$.

$1/\Gamma_{diff}$ can be determined using the following equation:

$$\frac{1}{\Gamma_{diff}} = \frac{0.75+0.238Kn}{Kn(1+Kn)} \ . \tag{2}$$

where Kn is Knudsen number. Knudsen number is defined as

$$Kn = \frac{\lambda}{a} \ , \tag{3}$$

where $\lambda$ is the mean free path of a molecule in the gas phase and a is the radius of the particle.

$\lambda$ can be derived from

$$\lambda = \frac{3D_g}{c}, \tag{4}$$

where $D_g$ is the diffusion coefficient in the gas phase and c is the mean molecular velocity.

c is determined by

$$c = \sqrt{\frac{8RT}{\pi M}} \tag{5}$$

where R is the gas constant, T is temperature, and M is the molecular mass of $SO_2$.

$1/\Gamma_{diff}$ was calculated to be 78 and $1/\gamma$ was calculated to be ~$8.3\times10^4$. $1/\Gamma_{diff}$ only accounted for <0.1% of $1/\gamma$. Therefore, the reactive uptake of $SO_2$ in this study was not limited by gas phase diffusion. The same conclusion can also be drawn by calculating the gas phase diffusion correction factor for a reactive uptake coefficient according to the method in Pöschl et al. (2007) (Equation 20 in their study, also shown as equation 6 below).

$$C_g = \frac{1}{1+\gamma\frac{0.75}{Kn}} \tag{6}$$

where $C_g$ is the gas phase diffusion correction factor for a reactive uptake coefficient."

*Was relative humidity actually measured or only inferred from mixing ratios of humid and dry air? RH is well know to be unpredictable in experiments, it should be measured directly.*

**Response:**

Relative humidity was measured directly in our study using a hygrometer (HMT100, Vaisala). In the revised manuscript, we will clearly describe this as follows.

"RH and temperature were measured using a hygrometer (HMT100, Vaisala) at the exit for gases of the reaction cell. Additionally, temperature was measured using another small temperature sensor (Pt 100, 1/3 DIN B; Heraeus) in the reaction cell."

*Line 36: This may be a language issue but it is not appropriate to refer to a point of disagreement in the literature which has prompted detailed analysis and publications as "different opinions." Replace*

*with "uncertainties in the pH value..." or something similar.*

**Response:**

Agree. In the revised manuscript, we will change "different opinions" to "uncertainties in the pH value".

*Line 56: Delete "$O_2$ is abundant in the atmosphere," this is an atmospheric chemistry journal.*

**Response:**

Agree. In the revised manuscript, we will delete this sentence.

**References**

Davidovits, P., Kolb, C. E., Williams, L. R., Jayne, J. T., and Worsnop, D. R.: Mass accommodation and chemical reactions at gas-liquid interfaces, Chem. Rev., 106, 1323-1354, 10.1021/cr040366k, 2006.

Liu, Y. J., Zhu, T., Zhao, D. F., and Zhang, Z. F.: Investigation of the hygroscopic properties of Ca(NO3)(2) and internally mixed Ca(NO3)(2)/CaCO3 particles by micro-Raman spectrometry, Atmos. Chem. Phys., 8, 7205-7215, 2008.

Pöschl, U., Rudich, Y., and Ammann, M.: Kinetic model framework for aerosol and cloud surface chemistry and gas-particle interactions - Part 1: General equations, parameters, and terminology, Atmos. Chem. Phys., 7, 5989-6023, 2007.

Zhao, D., Song, X., Zhu, T., Zhang, Z., and Liu, Y.: Multiphase Reaction of SO2 with NO2 on CaCO3 Particles. 1. Oxidation of SO2 by NO2, Atmos. Chem. Phys. Discuss., 2017, 1-23, 10.5194/acp-2017-610, 2017.

Zhao, D. F., Zhu, T., Chen, Q., Liu, Y. J., and Zhang, Z. F.: Raman micro-spectrometry as a technique for investigating heterogeneous reactions on individual atmospheric particles, Sci. China Chem., 54, 154-160, 10.1007/s11426-010-4182-x, 2011.

---

## Author Response (AR1)

**Response to reviewer #1**

We thank the reviewer for carefully reviewing our manuscript; the comments are greatly appreciated. We have addressed all the comments and made major changes to the manuscript. In particular, we have substantially revised the section of "Introduction", "Experimental", and "Reaction mechanism". We believe that the revisions based on these comments have substantially improved our manuscript. In the following please find our responses to the comments one by one, and the corresponding changes made to the manuscript. The original comments are shown in italics. The changes made in the revised manuscript are highlighted.

*General comments*

*In this manuscript, the results on multiphase reaction of $SO_2$ on individual $CaCO_3$ particles in the presence of $NO_2$ and $O_2$ at RH 72% using Micro-Raman spectrometer with a flow reaction system are presented. The main conclusion is that $CaCO_3$ converts first to $Ca(NO_3)_2$ droplet (by the reaction with $NO_2$), where actually further aqueous $SO_2$ oxidation by $O_2$ takes place. The reactive uptake coefficient γ of $SO_2$ determined on the basis of $SO_4^{2-}$ formation rate was ca. three orders of magnitude higher than that determined in the absence of $O_2$. On the basis of their results and mainly on literature data, they suggested that NO2 first initiates a free-radical chain mechanism via reaction of $NO_2$ with $HSO_3^-/SO_3^{2-}$, where $SO_3^{\bullet-}$ radical is formed, which reacts quickly with $O_2$ to form $SO_5^{\bullet-}$, etc…The presented mechanism is well known and confirmed by many authors, and as expected the reaction under $O_2$ conditions is relatively fast and can be important source of sulfate.*

*As I have already pointed out in the previous review (Zhao et al., ACPD), I support the topic, mainly due still unresolved questions concerning high sulfate formation shown during heavily polluted episodes under haze conditions in China. And also, I like the approach used for studying processes on the micro level.*

*However, I found many mistakes (not only English language, but in general); the article is not well readable and many times confusing, many sentences are repeated with no need through the manuscript; thus, I cannot recommend it in the present form for publication in ACP.*

*Further, I again strongly suggest presenting the results for both systems $SO_2/NO_2/H_2O/N_2$ and $SO_2/NO_2/H_2O/O_2$ together in one article, although the authors of the first manuscript gave their reasons (in the responses) why to present separately. I think it is reasonable to show both together, first due to easier comparison, further due to easier discussion on differences in the mechanisms, etc. Anyway, the authors compare the results with the*

*previous one during the whole manuscript. In addition, the experimental part is practically the same.*

*I highlight some of the main comments, questions and mistakes below. I will not expose the language mistakes, they are too many. Many parts of the manuscript should be rewritten.*

**Response:**

In the revised manuscript, we have made major changes to the manuscript in order to make the text more clearly written with improved readability. Moreover, we have made a thorough language check of the text of this manuscript and corrected the errors in language.

Regarding the arrangement of the two manuscripts, although they have some parts in common, the scientific questions under investigation and the chemistry in these two studies are substantially different. We have decided to arrange the two manuscripts as independent papers instead of companion papers to highlight their distinct features. The reasons for this arrangement are as follows:

1. The question whether the multiphase reaction of $SO_2$ directly with $NO_2$ is an important pathway for sulfate formation in the real atmosphere, e.g., during haze episodes in China, requires investigating the "pure" reaction of $SO_2$ with $NO_2$ without confounding effects of other oxidants. In another manuscript (Zhao et al., 2017), we address this question. And in this manuscript, we address a different question, i.e., whether the multiphase reaction of $SO_2$ with $O_2$ in the presence of $NO_2$ is an important reaction pathway of $SO_2$ oxidation; obviously, this reaction is more relevant to real atmospheric conditions.

2. We found that the multiphase reaction of $SO_2$ with $NO_2$ and the reaction of $SO_2$ with $O_2$ in the presence of $NO_2$ have significantly different chemistry, with different reaction mechanisms, products, and atmospheric implications.

i. The multiphase reaction of $SO_2$ directly with $NO_2$ involves two reactants whereas the reaction of $SO_2$ with $O_2/NO_2$ involves three reactants. In the former reaction $NO_2$ is the main oxidant of $SO_2$ while in the latter reaction $O_2$ is the main oxidant of $SO_2$ and $NO_2$ only acts as an initiator of chain reactions.

ii. According to the reaction mechanism, the main products in the multiphase reaction of $SO_2$ with $NO_2$ are sulfate and nitrite with a stoichiometry of 1:1. In contrast, the main product in the multiphase reaction of $SO_2$ with $O_2$ in the presence of $NO_2$ is sulfate and the ratio of sulfate to nitrite is expected to be much higher than 1:1 according to the free-radical chain mechanism in the present study because nitrite is only formed in the chain initiation step.

iii. Due to the different reaction mechanism, the atmospheric implication of the reaction of $SO_2$ with $O_2$ in the presence of $NO_2$ is significantly different from the direct reaction of $SO_2$

with $NO_2$ because the former leads to much faster sulfate formation.

Based on these reasons, we have changed the title of this manuscript to "$NO_2$-initiated Multiphase Oxidation of $SO_2$ by $O_2$ on $CaCO_3$ Particles" in the revised manuscript. Accordingly, in the revised manuscript we have deleted the parts where we stated about the companionship of these two papers (lines 63-64, 80 in the manuscript). We have also adjusted the wording in the abstract and introduction to reflect this change.

***Specific comments***

*Introduction**: It needs to be rewritten.*

*1. The authors should say something on well known and published mechanisms in the introduction. Discussion on the p.6/7 should partly be included here. Especially, the mechanism (R1-R6) does not fit on p.6, because it seems that is not important for their results.*

**Response:**

Agree. In the revised manuscript, we have made major changes to the introduction and added the published mechanism to the introduction.

As the reviewer suggested, in the introduction we have also included part of the discussion regarding the mechanism on the p.6-7. It now reads:

"Despite such a reaction mechanism for $SO_2$ oxidation being proposed, its role in $SO_2$ oxidation in the ambient atmosphere is not well established. Most previous studies have focused on the direct reaction of $SO_2$ with $NO_2$, including the determination of its rate constant (Lee and Schwartz, 1983; Clifton et al., 1988; Shen and Rochelle, 1998; Spindler et al., 2003; Nash, 1979; Huie and Neta, 1986). According to the reaction products and their reported yields (Lee and Schwartz, 1983; Clifton et al., 1988), the overall reaction equations of the direct reaction of $SO_2$ with $NO_2$ are as follows:

$$2NO_2(aq) + HSO_3^-(aq) + H_2O \rightarrow 2NO_2^-(aq) + SO_4^{2-}(aq) + 3H^+(aq), \quad \text{(R1)}$$

$$2NO_2(aq) + SO_3^{2-}(aq) + H_2O \rightarrow 2NO_2^-(aq) + SO_4^{2-}(aq) + 2H^+(aq), \quad \text{(R2)}$$

and the reactions are proposed to proceed via $NO_2$–S(IV) adduct complexes (Clifton et al., 1988).

$$NO_2(aq) + SO_3^{2-}(aq) \rightarrow [NO_2 - SO_3]^{2-}(aq). \quad \text{(R3)}$$

$$NO_2(aq) + [NO_2 - SO_3]^{2-}(aq) \rightarrow [NO_2 - SO_3 - NO_2]^{2-}(aq). \quad \text{(R4)}$$

$$[NO_2 - SO_3 - NO_2]^{2-}(aq) + OH^-(aq) \rightarrow [NO_2 - SO_4H - NO_2]^{3-}(aq).$$

$$\text{(R5)}$$

$$[NO_2 - SO_4H - NO_2]^{3-}(aq) \rightarrow 2NO_2^-(aq) + SO_4^{2-}(aq) + H^+(aq).$$
(R6)"

*2. P. 2, line 39 and p. 8, line 243: Dissolution/absorption of SO₂ in aqueous solution results in total dissolved S(IV), i.e. three species SO₂•H₂O, HSO₃⁻ and SO₃²⁻, which are in equilibrium; which species will prevail depends on pH! So, H₂SO₃ (2H⁺ +SO₃²⁻) are the same species as you have already written. In addition, it is not "rapid interconversion" between the species.*

**Response:**

Agree. In the revised manuscript, we have changed "$H_2SO_3$" to "$SO_2 \cdot H_2O$". And we have changed "rapid interconversion" to "fast dissociations of $SO_2 \cdot H_2O$ and $HSO_3^-$".

*Experimental:*

*3. RH is not mentioned in the experimental part, only in figure captions (RH 72%) and Table 1 (RH 82%). Is this fine or wrong?*

**Response:**

The values for RH in figures and Table 1 are correct. The experiments were carried out at two RH, 72% and 82%. The experiments with varied concentrations of $O_2$ were done at 82 % RH. For the experiments at 72% RH, the reactive uptake coefficients were not quantified. Therefore, only the reactive uptake coefficients at 82% RH are shown in Table 1. In the revised manuscript, we have elaborated the experimental part and clarified this point as follows.

"The experiments were conducted under two RHs (72% and 82%) at 75 ppm $SO_2$ and 75 ppm $NO_2$."

*4. Can you say something on pH of the formed aqueous layer of Ca(NO₃)₂? If you know the pH you can say something more on mechanism; i.e. if it is above 6, than SO₃²⁻ species are the main species which are involved.*

**Response:**

The pH of the aqueous layer of $Ca(NO_3)_2$ may not be completely homogeneous within the aqueous layer and may change dynamically with time during the reaction. In the surface of the aqueous layer pH was supposed to be lower, which was mainly determined by the gas-aqueous equilibrium of $SO_2$, and estimated to be ~3. In the vicinity of the $CaCO_3$ core, pH was supposed to be higher due to carbonate hydrolysis, and was estimated to be ~7.6.

Additionally, in the beginning of the reaction the overall pH of the aqueous layer should be higher due to the larger $CaCO_3$ core and thinner aqueous layer while in the end of the reaction overall pH should be lower, which was mainly determined by the gas-aqueous equilibrium of $SO_2$. Therefore, it is likely that both $HSO_3^-$ and $SO_3^{2-}$ were present, and the dominant species depended on the reaction time and location within the aqueous droplet. In the revised manuscript, we have added short discussion on this point.

"The dominant S(IV) species depends on pH. Due to the fast dissociations of $SO_2 \bullet H_2O$ and $HSO_3^-$, reactions consuming one of these S(IV) species will result in instantaneous re-establishment of the equilibria between them (Seinfeld and Pandis, 2006). In this study, the pH of the aqueous layer of $Ca(NO_3)_2$ may change dynamically with time during the reaction and may not be completely homogeneous within the aqueous droplet. The pH values could vary between ~3 and ~7.6. In the surface of the aqueous layer, pH was mainly determined by the gas–aqueous equilibrium of $SO_2$, and was estimated to be ~3. In the vicinity of the $CaCO_3$ core, pH was mainly determined by the hydrolysis of carbonate and was estimated to be ~7.6. It is likely that both $HSO_3^-$ and $SO_3^{2-}$ were present, and the dominant species depended on the reaction time and location within the aqueous droplet. Nevertheless, to make the reaction mechanism clearer, $HSO_3^-$ was used in the reaction equations. Similar reaction equations are also applicable to $SO_3^{2-}$ because of the fast dissociations of $SO_2 \bullet H_2O$ and $HSO_3^-$."

*Results and discussion:*

*5. If you once define that you have a system $NO_2/H_2O/O_2$, where $O_2$ is from synthetic air, there is no need to repeat "in synthetic air" throughout the manuscript.*

**Response:**

Agree. In the revised manuscript, we do not use "in synthetic air" throughout the manuscript where it is not necessary.

*6. Paragraph 3.2 An explanation on increasing concentration of $NO_3^-$ during the reaction is needed.*

**Response:**

Agree. The concentration of $NO_3^-$ increased during the reaction because $NO_3^-$ was continuously formed by the reaction of $CaCO_3$ with $NO_2$ and $H_2O$. In the revised manuscript, we have briefly discussed this point.

"The decrease in the amount of carbonate and the increase in the amount of nitrate was because carbonate reacted continuously with $NO_2$ and $H_2O$, forming $Ca(NO_3)_2$. The detailed

mechanism of the multiphase reaction of carbonate with $NO_2$ and $H_2O$ were discussed in our previous studies (Li et al., 2010; Zhao et al., 2017)."

*7. P.5, line 146: The reactive uptake coefficient γ of $SO_2$ was determined at three different $O_2$ concentrations, and not in the range 5-86%.*

**Response:**

Agree. In the revised manuscript, we have changed "with $NO_2$ in presence of $O_2$ (5%-86%)" to "with $O_2/NO_2$ at three $O_2$ concentrations (5, 20, and 86%)".

*8. P. 5, line 147: I can see that the increase in the reactive uptake coefficient γ of $SO_2$ is ca. three orders (and not two to three) of magnitude higher than that determined in the absence of $O_2$.*

**Response:**

The reactive uptake coefficient γ of $SO_2$ for the reaction with $O_2/NO_2$ in synthetic air ($1.2×10^{-5}$) was around 370 times higher than that determined for the direct oxidation of $SO_2$ by $NO_2$ ($3.22×10^{-8}$). Therefore, we described this difference as "two to three orders of magnitude".

*9. In the second paragraph of 3.3 you compare your results with the literature. Of course, that your results are different than that from Lee and Swartz (1983), due to many reasons, but probably the most important is their different approach. You can probably make some comparisons of your reaction rates with the rates got for the gas mixture $SO_2/NO_2/air$ introduced into aqueous solution (Turšič et al., 2001).*

**Response:**

We thank the reviewer for the suggestion. Regarding the experimental approach, Turšič et al. (2001) studied the absorption of the $SO_2/NO_2/air$ mixture into aqueous solution, which is indeed more similar to our study. However, due to the mass transfer limitations, it is difficult to directly compare the reaction rates in that study with ours and the rate constants in other studies. A rate constant of $2.4×10^3$ $mol^{-1}$ L $s^{-1}$ (at pH 3) can be derived from the study of Turšič et al. (2001) , which is much lower than the values from the study of Lee and Schwartz (1983) and of Clifton et al. (1988). This is likely attributed to the limiting step by the aqueous phase mass transfer. The characteristic mixing time in the aqueous phase in the study of Turšič et al. (2001) is likely much longer than that of Lee and Schwartz (1983) (1.7-5.3 s) according to the time series of $HSO_3^-$ concentration in the study of Turšič et al. (2001),

although it was not explicitly reported. Nevertheless, in the revised manuscript, we have added this comparison as follows.

"Only few studies have reported the S(IV) oxidation rate in the reaction of S(IV) with $O_2$/$NO_2$ mixtures (Turšič et al., 2001; Littlejohn et al., 1993). However, due to the limiting step by the aqueous phase mass transfer, it is difficult to quantitatively compare the reaction rates in those studies with the uptake coefficient in our study and the rate constants determined by Lee and Schwartz (1983) and Clifton et al. (1988). For example, a rate constant of $2.4 \times 10^3$ $mol^{-1}$ $L$ $s^{-1}$ (at pH 3) can be derived from the results of Turšič et al. (2001), which is much lower than the values reported by Lee and Schwartz (1983) and Clifton et al. (1988). This can be attributed to the limiting step by the aqueous-phase mass transfer because the characteristic mixing time in the aqueous phase in Turšič et al. (2001) was likely much longer than that of Lee and Schwartz (1983) (1.7–5.3 s), according to the $HSO_3^-$ concentration time series reported by Turšič et al. (2001)."

*10. The mechanism shown on p.6 (from line 172 to 189) should be deleted here and just mentioned in the introduction. It is not important for the discussion, but can be written in one sentence why this mechanism is not possible.*

**Response:**

Agree. In the revised manuscript, we have deleted the mechanism from line 172 to 189 and only mentioned them in the introduction, as the reviewer suggested. Instead, we have briefly discussed why this mechanism is not possible as follows.

"According to the $NO_2$–S(IV) adduct mechanism, the presence of $O_2$ should not affect the $SO_2$ oxidation rate; however, in this study, a substantial enhancement in the $SO_2$ oxidation rate was observed in the presence of $O_2$ compared with that in the absence of $O_2$. Therefore, the $NO_2$–S(IV) adduct mechanism was not considered to have been important in this study."

*11. Check the reactions in the text and in the Table 2 (see R7-R8 in the text and those in the Table, R8–R10, R8a, R8b)! Anyway, it is no need to repeat; all important reactions in the Table are enough.*

*If you know the pH, you can write the reaction with only one species, $HSO_3^-$ or $SO_3^{2-}$.*

**Response:**

Agree. In the revised manuscript, we have changed R8 to R7. We have deleted R7-17 in the text and only shown them in the table. Moreover, as discussed above (the response to comment 4) we have only shown the reaction equations for $HSO_3^-$ for clarity, although these reaction equations are also applicable to $SO_3^{2-}$ because of the fast dissociations of $SO_2 \cdot H_2O$

and $HSO_3^-$.

*12. P.7, line 209-213: Nice study on $S_2O_6^{2-}$ species formation, although in a different system, can be found in Podkrajšek et al, Chemosphere 49 (2002). Whatever, the mechanism (and the reason) of its formation should be the same.*

**Response:**

We thank the reviewer for raising this study on $S_2O_6^{2-}$ formation. In the revised manuscript, we have added this paper in our citation and briefly discussed it as follows.

"In addition to $SO_4^{2-}$ and $NO_2^-$, $S_2O_6^{2-}$ was detected with an appreciable yield using Raman spectroscopy, following the reaction of $NO_2$ with aqueous sulfite (Littlejohn et al., 1993). $S_2O_6^{2-}$ was also observed in the aqueous oxidation of bisulfite in an $N_2$-saturated solution in the presence of Fe(III) using ion-interaction chromatography (Podkrajšek et al., 2002)."

*13. Better than "radical mechanism" is "a free-radical chain mechanism".*

**Response:**

Agree. In the revised manuscript, we have changed the "radical mechanism" to "a free-radical chain mechanism".

*14. P. 8, Line 239: I do not agree that there is a synergy between $NO_2$ and $O_2$ (definition of synergy!); $NO_2$ only initiates the free-radical chain mechanism, and after the induction period, the reaction is relatively fast; and according to the proposed mechanism does not have other role, except in the first step when $Ca(NO_3)_2$ is formed in the reaction between $CaCO_3$ and $NO_2$. This part is now explained well in the first manuscript (Zhao et al., ACPD).*

**Response:**

By synergy, we meant that the overall effect on the $SO_2$ oxidation rate when both $NO_2$ and $O_2$ were present was higher than the sum of the effect of $NO_2$ and of $O_2$, although the reactions of $SO_2$ with $O_2$ and with $NO_2$ were not always simultaneous. As we have shown, the $SO_2$ oxidation rates in the direct reaction of $SO_2$ with $NO_2$ and with $O_2$ were both very low.

In the revised manuscript, we have further elaborated this discussion as follows.

" In the experiment without $NO_2$, but with other reaction conditions the same, we found that no sulfate was formed after 5 h of reaction. This indicates that $O_2$ by itself cannot initiate the chain reactions (although it favors chain propagation), and that the oxidation of $SO_2$ by $O_2$ was slow. The effect on the $SO_2$ oxidation rate when both $NO_2$ and $O_2$ were present was much higher than the sum of the effect of $NO_2$ and $O_2$. We refer to this effect as the synergy

of $NO_2$ and $O_2$, which resulted in the fast oxidation of $SO_2$ to form sulfate in this study. This effect is similar to a "ternary" reaction found with the reaction of $NO_2$–particles–$H_2O$ or $SO_2$–particles–$O_3$ (Zhu et al., 2011), where the reaction rate can be much faster than the sum of the reaction rates for the reaction of the second and third reactant with the first reactant."

*15. I also suggest excluding older references, and rather including only those after 1990.*

**Response:**

We thank the reviewer for the suggestion. However, some of the references earlier than 1990 are also important for the discussion of this study such as Lee and Schwartz (1983). Since ACP does not limit the number of references, we think it may be more reasonable to keep these references.

*16. The radicals throughout the manuscript are written incorrectly.*

**Response:**

Agree. In the revised manuscript, we have checked and corrected the writing of radicals throughout the manuscript wherever there were mistakes.

**References**

[revised manuscript text omitted]

**Response to reviewer #2**

We thank the reviewer for carefully reviewing our manuscript and providing helpful comments. All the comments have been addressed in the revised manuscript and we believe that the revisions based on these comments have substantially improved our manuscript. In the following please find our responses to the comments one by one, and the corresponding changes made to the manuscript. The original comments are shown in italics. The changes made in the revised manuscript are highlighted.

*I support the other referee's assessment, including the recommendation that Part 1 and Part 2 be combined into a single article. The articles don't stand on their own, and studying the reaction of $SO_2$ with $NO_2$ in the absence of $O_2$, as in Part 1 - since $O_2$ apparently plays a role in the reaction - is not relevant for atmospheric chemistry. I also agree that there are numerous English language errors in this manuscript.*

**Response:**

We appreciate the reviewer's opinion on how to better present our studies. However, these two studies are substantially different regarding the scientific questions under investigation and the chemistry involved, although they have some links. We have decided to arrange the two manuscripts as independent papers instead of companion papers to highlight their distinct features. The reasons for this arrangement are as follows:

1. The question whether the multiphase reaction of $SO_2$ directly with $NO_2$ is an important pathway for sulfate formation in the real atmosphere, e.g., during haze episodes in China, requires investigating the "pure" reaction of $SO_2$ with $NO_2$ without confounding effects of other oxidants. In another manuscript (Zhao et al., 2017), we address this question. And in this manuscript, we address a different question, i.e., whether the multiphase reaction of $SO_2$ with $O_2$ in the presence of $NO_2$ is an important reaction pathway of $SO_2$ oxidation. Both reactions have their own relevance to the atmosphere.

2. We found that the multiphase reaction of $SO_2$ with $NO_2$ and the reaction of $SO_2$ with $O_2$ in the presence of $NO_2$ have significantly different chemistry, with different reaction mechanisms, products, and atmospheric implications.

i. The multiphase reaction of $SO_2$ directly with $NO_2$ involves two reactants whereas the reaction of $SO_2$ with $O_2/NO_2$ involves three reactants. In the former reaction $NO_2$ is the main oxidant of $SO_2$ while in the latter reaction $O_2$ is the main oxidant of $SO_2$ and $NO_2$ only acts as an initiator of chain reactions.

ii. According to the reaction mechanism, the main products in the multiphase reaction of $SO_2$ with $NO_2$ are sulfate and nitrite with a stoichiometry of 1:1. In contrast, the main product in

the multiphase reaction of $SO_2$ with $O_2$ in the presence of $NO_2$ is sulfate and the ratio of sulfate to nitrite is expected to be much higher than 1:1 according to the free-radical chain mechanism in the present study because nitrite is only formed in the chain initiation step.

iii. Due to the different reaction mechanism, the atmospheric implication of the reaction of $SO_2$ with $O_2$ in the presence of $NO_2$ is significantly different from the direct reaction of $SO_2$ with $NO_2$ because the former leads to much faster sulfate formation.

Based on these reasons, we have changed the title of this manuscript to "$NO_2$-initiated Multiphase Oxidation of $SO_2$ by $O_2$ on $CaCO_3$ Particles" in the revised manuscript. Accordingly, in the revised manuscript we have deleted the parts where we stated about the companionship of these two papers (lines 63-64, 80 in the manuscript). We have also adjusted the wording in the abstract and introduction to reflect this change.

Regarding the language errors, we have made a thorough language check of the text of this manuscript and corrected the errors in language.

*The study and the results presented are interesting. However, I have doubts about the technical soundness of the approach. For one thing, it's impossible to understand the experimental approach based on what is written in section 2 of this manuscript. Yes, the reactor was a flow reactor, but where were the particles? Were they part of the flow? Or were gases flowing past the particles which are stationary on a surface? What were the particles like (size, shape, porosity, etc.)? How were they prepared and dispersed? The manuscript is meaningless without these basic pieces of information. It is possible that they were mentioned in Part I of the manuscript, but I am being asked to review only this manuscript - and it must stand on its own at least to this extent.*

**Response:**

Agree. In the revised manuscript, we have elaborated the experimental part. The details of experiments including the details that the reviewer is concerned with have been added and explicitly described. The revision has also reflected the change from the companion paper of our last manuscript to a completely independent paper. The revised experimental part is as follows.

[revised manuscript text omitted]

*I suspect that the particles were not part of the flow. In which case, did the authors consider the issue of gas phase diffusion limitations in their data analysis?*

**Response:**

Particles were deposited on a Teflon-FEP film in this study. Please refer the description about the experimental setup.

We evaluated the influence of gas phase diffusion on the reactive uptake coefficient using the resistor model described by Davidovits et al. (2006) and references therein (see details in the

Supplement S1 below, which has been added in the revised manuscript). The contribution of the resistance ($1/\Gamma_{diff}$) due to gas phase diffusion to the reactive uptake coefficient in this study was less than 0.1%. Therefore we conclude that the reactive uptake of $SO_2$ was not limited by gas phase diffusion. The same conclusion can also be drawn by calculating the gas phase diffusion correction factor for a reactive uptake coefficient according to the method in Pöschl et al. (2007) (Equation 20 in their study).

In the revised manuscript, we have briefly discussed this point.

"The influence of gas phase diffusion on reactive uptake was evaluated using the resistor model described by Davidovits et al. (2006) and references therein, as well as using the gas phase diffusion correction factor for a reactive uptake coefficient according to the method described by Pöschl et al. (2007). The reactive uptake of $SO_2$ was found to not be limited by gas phase diffusion (see details in the Supplement S1)."

The Supplement S1 that has been added in the revised manuscript is as follows.

"**S1. Influence of gas phase diffusion on reactive uptake coefficients**

The Influence of the gas phase diffusion on reactive uptake coefficients was analyzed using the resistor model described by Davidovits et al. (2006) and the references therein.

$$\frac{1}{\gamma} = \frac{1}{\Gamma_{diff}} + \frac{1}{\alpha} + \frac{1}{\Gamma_{sat} + \Gamma_{rxn}} \qquad (1)$$

where $\Gamma_{diff}$ is the transport coefficient in the gas phase, $1/\Gamma_{diff}$ is the resistance due to the diffusion in the gas phase. Similarly, $1/\Gamma_{sat}$ and $1/\Gamma_{rxn}$ are the resistance due to liquid phase saturation and liquid phase reaction, respectively. $\alpha$ is the mass accommodation coefficient of $SO_2$.

$1/\Gamma_{diff}$ can be determined using the following equation:

$$\frac{1}{\Gamma_{diff}} = \frac{0.75 + 0.238 Kn}{Kn(1 + Kn)} \cdot \qquad (2)$$

where Kn is Knudsen number. Knudsen number is defined as

$$Kn = \frac{\lambda}{a} , \qquad (3)$$

where $\lambda$ is the mean free path of a molecule in the gas phase and a is the radius of the particle. $\lambda$ can be derived from

$$\lambda = \frac{3D_g}{c}, \qquad (4)$$

where $D_g$ is the diffusion coefficient in the gas phase and c is the mean molecular velocity. c is determined by

$$c = \sqrt{\frac{8RT}{\pi M}} \qquad (5)$$

where R is the gas constant, T is temperature, and M is the molecular mass of $SO_2$.

$1/\Gamma_{diff}$ was calculated to be 78 and $1/\gamma$ was calculated to be $\sim 8.3 \times 10^4$. $1/\Gamma_{diff}$ only accounted for <0.1% of $1/\gamma$. Therefore, the reactive uptake of $SO_2$ in this study was not limited by gas phase diffusion.

The same conclusion can also be drawn by calculating the gas phase diffusion correction factor for a reactive uptake coefficient according to the method in Pöschl et al. (2007) (Equation 20 in their study, also shown as equation 6 below).

$$C_g = \frac{1}{1+\gamma\frac{0.75}{Kn}} \tag{6}$$

where $C_g$ is the gas phase diffusion correction factor for a reactive uptake coefficient."

*Was relative humidity actually measured or only inferred from mixing ratios of humid and dry air? RH is well know to be unpredictable in experiments, it should be measured directly.*

**Response:**

Relative humidity was measured directly in our study using a hygrometer (HMT100, Vaisala). In the revised manuscript, we have clearly described this as follows.

"RH and temperature were measured using a hygrometer (HMT100, Vaisala, Vantaa, Finland) at the exit of the reaction cell. Additionally, temperature was measured using another small temperature sensor (Pt 100, 1/3 DIN B, Heraeus, Hanau, Germany) in the reaction cell."

*Line 36: This may be a language issue but it is not appropriate to refer to a point of disagreement in the literature which has prompted detailed analysis and publications as "different opinions." Replace with "uncertainties in the pH value..." or something similar.*

**Response:**

Agree. In the revised manuscript, we have changed "different opinions" to "uncertainties in the pH value".

*Line 56: Delete "O$_2$ is abundant in the atmosphere," this is an atmospheric chemistry journal.*

**Response:**

Agree. In the revised manuscript, we have deleted this sentence.

Although the direct oxidation of $SO_2$ by $NO_2$ only accounted for a very small fraction of sulfate formation, $NO_2$ played an important role in the $SO_2$ oxidation by initiating the chain reactions via the production of the $SO_3^{\cdot-}$ radical (R7). In the experiment without $NO_2$, but with other reaction conditions the same, we were unable to detect sulfate after 5 h of reaction. This indicates that $O_2$ by itself cannot initiate the chain reaction (although it favors chain propagation), and that the oxidation of $SO_2$ by $O_2$ was slow. The effect on the $SO_2$ oxidation rate when both $NO_2$ and $O_2$ were present was much higher than the sum of the effect of $NO_2$ and $O_2$. We refer to this effect as the synergy of $NO_2$ and $O_2$, which resulted in the fast oxidation of $SO_2$ to form sulfate in this study. This effect is similar to a "ternary" reaction found with the reaction of $NO_2$–particles–$H_2O$ or $SO_2$–particles–$O_3$ (Zhu et al., 2011), where the reaction rate can be much faster than the sum of the reaction rates for the reaction of the second and third reactant with the first reactant. In addition to acting as the initiator of chain reactions, $NO_2$ also contributed to the formation of the aqueous phase through the reaction with $CaCO_3$, forming $Ca(NO_3)_2$ as discussed above, which provided a site for S(IV) oxidation.

Based on the discussion above, we summarize the reaction mechanism that occurred in this study in Table 2. The reactions are classified as chain initiation, chain propagation, and chain termination. The dominant S(IV)

species depends on pH.  Due to the fast dissociations of $SO_2 \cdot H_2O$ and $HSO_3^-$, reactions consuming one of these S(IV) species will result in instantaneous re-establishment of the equilibria between them (Seinfeld and Pandis, 2006). In this study, the pH of the aqueous layer of $Ca(NO_3)_2$ may change dynamically with time during the reaction and may not be completely homogeneous within the aqueous droplet. The pH values could vary between ~3 and ~7.6. In the surface of the aqueous layer, pH was mainly determined by the gas–aqueous equilibrium of $SO_2$, and was estimated to be ~3. In the vicinity of the $CaCO_3$ core, pH was mainly determined by the hydrolysis of carbonate and was estimated to be ~7.6. It is likely that both $HSO_3^-$ and $SO_3^{2-}$ were present, and the dominant species depended on the reaction time and location within the aqueous droplet. Nevertheless, to make the reaction mechanism clearer, $HSO_3^-$ was used in the reaction equations. Similar reaction equations are also applicable to $SO_3^{2-}$ because of the fast dissociations of $SO_2 \cdot H_2O$ and $HSO_3^-$. Overall, the reaction can be written as follows, which  clearly shows that $O_2$ was the main oxidant for sulfate formation:

$$2NO_2(aq) + 2HSO_3^-(aq) + (0/1)O_2 \rightarrow 2NO_2^-(aq) + S_2O_{6/8}^{2-}(aq) +$$
$$2H^+(aq). \qquad (R18)$$

$$\sim\sim 2SO_3^{2-}(aq) + O_2 \rightarrow 2SnO_2 + 2nHSO_3^-(aq) \rightarrow 2nSO_4^{2-}(aq)(aq) + 2nH^+(aq),$$
$$(R19)$$

[revised manuscript text omitted]

---

## Referee Report (RR1)

**General:** I am receiving the revised manuscript without having reviewed it before. I think the paper is generally ok and the authors have addressed the concerns of the reviewers.

The authors tend to see the observed oxygen-dependence as a support for the direct single electron transfer between HSO3- and NO2 which directly leads to SO3- and NO2- / HONO. The finding of the oxygen effect is interesting but possible more measured points would have been helpful – that could be included into an outlook. Please also note that the NO2 and SO2 mixing ratios applied are gigantic.

Regarding the mechanism of the reaction between S(IV) and N(IV) thermochemistry should be considered. Such treatment shows that the reaction is only exothermic by a very small extend and it should be slow. The complex formation mechanism appears much more feasible to me because of this. Maybe oxygen can promote the decomposition of the adduct by scavenging the formed SO3-, if there is not only a unimolecular decomposition of the complex but an equilibrium. Then oxygen would be able to promote scavenging of SO3- to SO5- thus shifting a decomposition equilibrium on the product side. Maybe this could be discussed and included into the discussion / outlook.

Anyway, whether there is a concerted single electron transfer or whether the reaction runs through a complex, at the end sulfur-oxy radical anions will be produced the role of which is S(IV) oxidation has been studied extensively in the late 1980s and early 1990 both in the US and in Europe. These reactions are included into state-of-the art aqueous chemical mechanism and a chemistry such as suggested here would add a source of SO3-. One thought which is important for the whole scope of th e paper: Such an additional SO3- radical anion source will not lead to a much increased S(VI) production as the sulfur radical chain will not establish but the reactive chain carriers will be scavenged and reduce the effectivity of the potential radical reaction chain. This has been demonstrate in many studies and , to some extent, it is surprising  that this is not being discussed adequately in a study motivated to explain particle sulfate formation correlating to NO2 in the gas phase. Especially in China, the sulfur-oxy radical chemistry will be scavenged by the abundance of organics in particles – this occurs via the reaction sof SO4- with organics where many kinetic data are available but also by such reactions of SO5- where the data basis is more sparse. This overall thought should surely be considered in the study.

**Details**

Line 53:        The postulation of complexes involving more than one NO2 unit probably comes from pulse radiolysis experiments with quite high NO2 concentrations. It should be considered that not only the complex with three NO2 unit can decompose but alos the others, especially the most simple on being formed in R3.

L 205-208:     See above remarks, I would be a bit more cautious here.

L 209-257:     All of this must be put into context with the sensitivity of the sulfur-oxy-radical reaction chain towards organics under real environmental conditions.

L  286 ff:      Conclusions section: I feel this should be modified somewhat in view of the above.

---

## Author Response (AR2)

**Response to referee #1**

We thank the reviewer for further reviewing our manuscript and providing helpful comments. We have addressed these comments. In the following please find our response and the corresponding change made to the manuscript. The original comments are shown in italics. The changes made in the revised manuscript are highlighted.

*I feel that the authors have addressed my most serious concerns from the previous version. However, before the article is suitable for publication, the authors need to address the fact that these experiments were conducted at extremely high, environmentally unrealistic reactant concentrations (75 ppm $SO_2$ and $NO_2$). This is particularly relevant since the reaction products are expected to alter the properties of the particle surface (pH, hygroscopicity) and this process will occur unrealistically fast at elevated reactant concentrations.*

**Response:**

We have accepted the suggestion. In our experiments, the $SO_2$ and $NO_2$ concentrations much higher than those in ambient levels were used, because 1) enough sulfate as a reaction product can be formed on a single particle to obtain high signal-to-noise ratio in our experimental setup; 2) in laboratory studies, high concentrations of reactants are often used in order to simulate the chemical/physical processes within minutes or hours which occur in the ambient atmosphere on the time scale of days or weeks.

In the revised manuscript, we have discussed this limitation as follows.

"It is important to note that the concentrations of $NO_2$ and $SO_2$ used in this study are much higher than those in the ambient atmosphere. High concentrations of reactant gases are often used in laboratory studies in order to simulate the ambient reactions at the time scale of days or weeks and to get high signal-to-noise ratios for detecting products within minutes or hours. In the ambient atmosphere, reactive uptake coefficient of $SO_2$ should be lower than that in this study due to the lower $NO_2$ concentrations when other conditions are comparable and the chemical/physical processes observed in this study, such as changes in particle composition, phase, hygroscopicity, and pH should be much slower due to the lower concentrations of $NO_2$ and $SO_2$."

**Response to referee #2**

We thank the reviewer for carefully reviewing our manuscript and providing helpful comments. All the comments have been addressed in the revised manuscript and we believe that the revisions based on these comments have improved our manuscript significantly. In the following please find our responses to the comments one by one, and the corresponding changes made to the manuscript. The original comments are shown in italics. The changes made in the revised manuscript are highlighted.

*General: I am receiving the revised manuscript without having reviewed it before. I think the paper is generally ok and the authors have addressed the concerns of the reviewers.*

*The authors tend to see the observed oxygen-dependence as a support for the direct single electron transfer between $HSO_3^-$ and $NO_2$ which directly leads to $SO_3^-$ and $NO_2^-$ / HONO. The finding of the oxygen effect is interesting but possible more measured points would have been helpful – that could be included into an outlook. Please also note that the $NO_2$ and $SO_2$ mixing ratios applied are gigantic. Regarding the mechanism of the reaction between S(IV) and N(IV) thermochemistry should be considered. Such treatment shows that the reaction is only exothermic by a very small extend and it should be slow. The complex formation mechanism appears much more feasible to me because of this. Maybe oxygen can promote the decomposition of the adduct by scavenging the formed $SO_3^-$, if there is not only a unimolecular decomposition of the complex but an equilibrium. Then oxygen would be able to promote scavenging of $SO_3^-$ to $SO_5^-$ thus shifting a decomposition equilibrium on the product side. Maybe this could be discussed and included into the discussion / outlook.*

*Anyway, whether there is a concerted single electron transfer or whether the reaction runs through a complex, at the end sulfur-oxy radical anions will be produced the role of which is S(IV) oxidation has been studied extensively in the late 1980s and early 1990 both in the US and in Europe. These reactions are included into state-of-the art aqueous chemical mechanism and a chemistry such as suggested here would add a source of $SO_3^-$. One thought which is important for the whole scope of the paper: Such an additional $SO_3^-$ radical anion source will not lead to a much increased S(VI) production as the sulfur radical chain will not establish but the reactive chain carriers will be scavenged and reduce the effectivity of the potential radical reaction chain. This has been demonstrate in many studies and , to some extent, it is surprising that this is not being discussed adequately in a study motivated to explain particle sulfate formation correlating to $NO_2$ in the gas phase. Especially in China, the sulfur-oxy radical chemistry will be scavenged by the abundance of*

*organics in particles – this occurs via the reaction of $SO_4^-$ with organics where many kinetic data are available but also by such reactions of $SO_5^-$ where the data basis is more sparse. This overall thought should surely be considered in the study.*

**Response:**

We thank the reviewer for supportive remarks.

The effect of oxygen was shown to be significant. We agree with the reviewer that more oxygen levels would be helpful. In the revised manuscript, we have added a brief outlook to have experimental results with more oxygen levels as follows.

"In this study, we investigated the reaction of $SO_2$ with $O_2$ in the presence of $NO_2$ at three $O_2$ concentrations. The influence of the $O_2$ concentration was shown to be significant. Future experimental results with more $O_2$ concentration levels would provide more insights into the reaction mechanism and process."

Moreover, we have discussed the limitation of using high mixing ratios of $SO_2$ and $NO_2$ in the revised manuscript as follows.

"It is important to note that the concentrations of $NO_2$ and $SO_2$ used in this study are much higher than those in the ambient atmosphere. High concentrations of reactant gases are often used in laboratory studies in order to simulate the ambient reactions at the time scale of days or weeks and to get high signal-to-noise ratios for detecting products within minutes or hours. In the ambient atmosphere, reactive uptake coefficient of $SO_2$ should be lower than that in this study due to the lower $NO_2$ concentrations when other conditions are comparable and the chemical/physical processes observed in this study, such as changes in particle composition, phase, hygroscopicity, and pH should be much slower due to the lower concentrations of $NO_2$ and $SO_2$."

We agree that the mechanism involving the decomposition of the reaction complex and the equilibrium as the reviewer suggested is possible. In the revised manuscript, we have added discussion on this mechanism as follows.

"In addition to the two mechanisms above, Spindler et al. (2003) proposed a reaction mechanism involving first $NO_2$–S(IV) complex formation and subsequent $SO_3^{\bullet-}$ radical formation (R3, R7). $NO_2$–S(IV) complex may establish an equilibrium with $SO_3^{\bullet-}$ in contrast to the direct formation of $SO_3^{\bullet-}$ via the reaction of $NO_2$ with $SO_2$. Higher concentration of $O_2$ favors the conversion of $SO_3^{\bullet-}$ to $SO_5^{\bullet-}$ and thus can reduce the $SO_3^{\bullet-}$ concentration, shifting the equilibrium to the product side and promoting the overall S(IV) oxidation. $O_2$ can act in a similar way as in the free-radical chain mechanism. Admittedly, we cannot rule out the possibility $NO_2$–S(IV) complex formation. But such a mechanism can still be classified as the free-radical chain mechanism since the S(IV) oxidation still proceeds via the radical chain reactions."

We agree that in the ambient atmosphere in the internally mixed particles where organics and SI(IV) co-exist, organics in particles can scavenge sulfur-oxy radical anions and thus reduce the effectivity of the potential radical reaction chain and of S(IV) oxidation. In the revised manuscript, we have discussed this point as follows.

"In addition, in the ambient atmosphere, the internal mixing of organics with S(IV) in particles may influence the S(IV) oxidation rate by $O_2$ in the presence of $NO_2$. When organics is abundant in particles, for example during haze episodes in China, it can react with and thus scavenge radical anion carriers such as $SO_5^{\bullet-}$ and $SO_4^{\bullet-}$ (Herrmann, 2003; Herrmann et al., 2015; Huie, 1995). Therefore, the presence of internally mixed organics can reduce the effectivity of the potential radical reaction chain and of S(IV) oxidation, which can undermine the importance of the oxidation by $O_2$ in the presence of $NO_2$ in the overall S(IV) oxidation."

*Details*

*Line 53: The postulation of complexes involving more than one $NO_2$ unit probably comes from pulse radiolysis experiments with quite high $NO_2$ concentrations. It should be considered that not only the complex with three $NO_2$ unit can decompose but also the others, especially the most simple on being formed in R3.*

**Response:**

Accepted. In the revised manuscript, we have added the reaction equation of the decomposition of simple $SO_2$-$NO_2$ complex.

"Additionally, $NO_2$–S(IV) adduct complex may decompose as follows (Spindler et al., 2003).

$$[NO_2 - SO_3]^{2-}(aq) \rightarrow NO_2^-(aq) + SO_3^{\bullet-}(aq). \hspace{2cm} (R7)"$$

*L 205-208: See above remarks, I would be a bit more cautious here.*

**Response:**

Accepted. In the revised manuscript, we have revised this part.

"According to the $NO_2$–S(IV) complex mechanism, the presence of $O_2$ should not affect the $SO_2$ oxidation rate; however, in this study, a substantial enhancement in the $SO_2$ oxidation rate was observed in the presence of $O_2$ compared with that in the absence of $O_2$. Therefore, the $NO_2$–S(IV) complex mechanism was less likely to have been important in this study."

We further discussed the mechanism involving both $NO_2$–S(IV) complex and $SO_3^{\bullet-}$ radical as mentioned above.

"In addition to the two mechanisms above, Spindler et al. (2003) proposed a reaction mechanism involving first $NO_2$–S(IV) complex formation and subsequent $SO_3^{\bullet-}$ radical formation (R3, R7). $NO_2$–S(IV) complex may establish an equilibrium with $SO_3^{\bullet-}$ in contrast to the direct formation of $SO_3^{\bullet-}$ via the reaction of $NO_2$ with $SO_2$. Higher concentration of $O_2$ favors the conversion of $SO_3^{\bullet-}$ to $SO_5^{\bullet-}$ and thus can reduce the $SO_3^{\bullet-}$ concentration, shifting the equilibrium to the product side and promoting the overall S(IV) oxidation. $O_2$ can act in a similar way as in the free-radical chain mechanism. Admittedly, we cannot rule out the possibility $NO_2$–S(IV) complex formation. But such a mechanism can still be classified as the free-radical chain mechanism since the S(IV) oxidation still proceeds via the radical chain reactions."

*L 209-257: All of this must be put into context with the sensitivity of the sulfur-oxy-radical reaction chain towards organics under real environmental conditions.*

**Response:**

Accepted. In the revised manuscript, we have discussed the influence of organics in the particle phase on the sulfur-oxy-radical chain reactions as in the response to the general comments above.

*L 286 ff: Conclusions section: I feel this should be modified somewhat in view of the above.*

**Response:**

Accepted. In the revised manuscript, we have modified this section by adding the following text.

[revised manuscript text omitted]

$$\frac{1}{\gamma} = \frac{1}{\Gamma_{diff}} + \frac{1}{\alpha} + \frac{1}{\Gamma_{sat}+\Gamma_{rxn}} \tag{1}$$

where $\Gamma_{diff}$ is the transport coefficient in the gas phase, $1/\Gamma_{diff}$ is the resistance due to the diffusion in the gas phase. Similarly, $1/\Gamma_{sat}$ and $1/\Gamma_{rxn}$ are the resistance due to liquid phase saturation and liquid phase reaction, respectively. $\alpha$ is the mass accommodation coefficient of $SO_2$.

$1/\Gamma_{diff}$ can be determined using the following equation:

$$\frac{1}{\Gamma_{diff}} = \frac{0.75+0.238Kn}{Kn(1+Kn)} \; . \tag{2}$$

where Kn is Knudsen number. Knudsen number is defined as

$$Kn = \frac{\lambda}{a} \; , \tag{3}$$

where $\lambda$ is the mean free path of a molecule in the gas phase and a is the radius of the particle.

$\lambda$ can be derived from

$$\lambda = \frac{3D_g}{c}, \tag{4}$$

where $D_g$ is the diffusion coefficient in the gas phase and c is the mean molecular velocity.

c is determined by

$$c = \sqrt{\frac{8RT}{\pi M}} \tag{5}$$

where R is the gas constant, T is temperature, and M is the molecular mass of $SO_2$.

$1/\Gamma_{diff}$ was calculated to be 78 and $1/\gamma$ was calculated to be $\sim 8.3 \times 10^4$. $1/\Gamma_{diff}$ only accounted for <0.1% of $1/\gamma$. Therefore, the reactive uptake of $SO_2$ in this study was not limited by gas phase diffusion.

The same conclusion can also be drawn by calculating the gas phase diffusion correction factor for a reactive uptake coefficient according to the method in Pöschl et al. (2007) (Equation 20 in their study, also shown as equation 6 below).

$$C_g = \frac{1}{1+\gamma\frac{0.75}{Kn}} \tag{6}$$

where $C_g$ is the gas phase diffusion correction factor for a reactive uptake coefficient.

[Figure]

Figure S1. Schematic diagram of the experimental setup (Zhao et al., 2017). MFC: mass flow controller.

[Figure]

Figure S2. X-ray diffraction spectra of CaCO₃ particles.